# Strategic Classification with Non-Linear Classifiers

**Benyamin Trachtenberg,   Nir Rosenfeld**
Technion – Israel Institute of Technology
Haifa, Israel
{benyamint, nirr} @cs.technion.ac.il

## Abstract

In strategic classification, the standard supervised learning setting is extended to support the notion of strategic user behavior in the form of costly feature manipulations made in response to a classifier. While standard learning supports a broad range of model classes, the study of strategic classification has, so far, been dedicated mostly to linear classifiers. This work aims to expand the horizon by exploring how strategic behavior manifests under non-linear classifiers and what this implies for learning. We take a bottom-up approach showing how non-linearity affects decision boundary points, classifier expressivity, and model class complexity. Our results show how, unlike the linear case, strategic behavior may either increase or decrease effective class complexity, and that the complexity decrease may be arbitrarily large. Another key finding is that universal approximators (e.g., neural nets) are no longer universal once the environment is strategic. We demonstrate empirically how this can create performance gaps even on an unrestricted model class.

## 1   Introduction

Strategic classification considers learning in a setting where users modify their features in response to a deployed classifier to obtain positive predictions [3, 4, 16]. This setting aims to capture a natural tension that can arise between a system aiming to maximize accuracy and its users who seek favorable outcomes. Such a tension is well-motivated across diverse applications, including loan approval, university admission, job hiring, welfare and education programs, and medical services. The field has drawn much attention since its introduction, leading to advances along multiple learning fronts such as generalization [6, 33, 36], optimization [25], loss functions [24, 26], and extensions [11, 14, 18, 30].

However, despite over a decade of effort, research in strategic classification remains predominantly focused on linear classification. Due to the inherent challenges of strategic learning, this focus has been reasonable, given that linearity has the benefit of inducing a simple form of strategic behavior that permits tractable analysis. Yet, since many realistic applications—including those mentioned above—make regular use of non-linear models, we believe that it is important to establish a better understanding of how non-linearity shapes strategic behavior, and how behavior in turn affects learning. Our paper therefore aims to extend the study of strategic classification to the non-linear regime.

Our first observation is that non-linear classifiers induce qualitatively different behavior than linear ones in the strategic setting. Consider first the structure induced by standard linear classifiers. Generally, users in strategic classification are assumed to best-respond to the classifier by modifying their features in a way that maximizes utility (from prediction) minus costs (from modification). For a linear classifier $h$ coupled with the common choice of an $\ell_2$ cost, modifications $x \mapsto x^h$ manifest as a projection of $x$ onto the linear decision boundary of $h$ as long as it incurs less than a unit cost. This simple form has several implications. First, note that all points 'move' in the same direction, making strategic behavior, in effect, one-dimensional. Second, the region of points that move forms a band of unit distance on the negative side of $h$. This means that the set of points classified as positive

forms a halfspace, which implies that the 'effective classifier', given by $h_\Delta(x) := h(x^h)$, is also linear. Third, and as a result, the induced effective model class remains linear.

Together, the above suggest that for linear classifiers coupled with $\ell_2$ costs, the transition from the standard, non-strategic setting to a strategic one requires anticipating which points move, but otherwise remains similar. One known implication is that the strategic and non-strategic VC dimension of linear classifiers are the same [33]. Furthermore, the Bayes-optimal strategic classifier can be derived by taking the optimal non-strategic classifier and simply increasing its bias term by one [29]. Note that this means that any linearly separable problem remains separable, and that strategic classifiers can, in principle, attain the same level of accuracy as their non-strategic counterparts. Thus, while perhaps challenging to solve in practice, the intrinsic difficulty of strategic linear classification remains unchanged.

The above, however, is unlikely to hold for non-linear classifiers: once the decision boundary is non-linear, different points will move in different directions. This lends to the possible distortion of the shape of a given classifier, which in turn can change the effective (i.e., strategic) model class. Such changes can have concrete implications on expressivity, generalization, optimization, and attainable accuracy. This motivates the need to explore how strategic responses shape outcomes in non-linear strategic classification, which we believe is crucial as a first step towards developing future methods. Our general approach is to examine—through analysis, examples, and experiments—how objects in the original non-strategic task map to corresponding objects in the induced strategic problem instance. Initially, our conjecture was that strategic behavior acts as a 'smoothing' operator on the original decision boundary, but as we show, the actual effect of strategic behavior turns out to be more involved.

To build an intuition of the mechanisms at play, we take a bottom-up approach and study three mappings between (i) individual points, (ii) classifiers, and (iii) model classes. For points, we begin by showing that the mapping is not necessarily bijective. As a result, some points on the original decision boundary become 'wiped out', while others are 'expanded' to become intervals (arcs for $\ell_2$ costs) on the induced boundary. We demonstrate several mechanisms that can give rise to both phenomena. For classifiers, we show how the effects on individual points aggregate to modify the entire decision boundary. A key finding is that certain decision boundaries are *impossible to express* under strategic behavior. We give several examples of reasonable classifier types that cannot exist under strategic behavior, and show that some effective classifiers $h_\Delta$ can be traced back to (infinitely) many original classifiers $h$. The main takeaway is that universal approximators, such as neural networks or RBF kernels, are no longer universal in a strategic environment.

Our main contribution lies in our analysis of model classes, where our results build on the observation that the mapping from original to effective classifiers *within* a class is not necessarily bijective. In particular, because many $h$ can be mapped to the same $h_\Delta$, the VC dimension of the induced model class can shrink even from $\infty$ down to 0, causing model classes that were not learnable in the standard setting to become learnable in the strategic one. That said, the VC dimension of the induced class can generally either grow or shrink. We highlight some conditions that guarantee either of these outcomes, and, in particular, provide an upper bound on the strategic complexity increase for the expressive family of piecewise-linear model classes (e.g., ReLU networks). Our analysis yields several practical takeaways for learning, model selection, and future method development (see Appendix C.10). We conclude with experiments that shed light on how strategic behavior may alter the effective complexity and maximum accuracy of the learned model class in practice.

## 2   Related work

Our work focuses on the common setting of supervised binary classification, where non-linear models are highly prevalent in general. This follows the original formulation of strategic classification [3, 16], which has since been extended also to online settings [1, 5, 8] and regression [2, 31]. As noted, linear models have thus far been at the forefront of the field, with many works relying (explicitly or implicitly) on their properties or induced structure [5, 10, 18, 25, 29]. Even for works with results that are agnostic to model class selection, linear classifiers remain the focus of special-case analyses, examples, and empirical evaluation [20, 26, 37], and their generality does not shed light on the impact of non-linearity. Other works offer only tangential results, such as welfare analysis in the non-linear setting [35], but do not paint a picture of the general learning problem. Our low-level analysis makes use of ideas from the field of offset curves from computational geometry [12, 21]. Whereas this field is mostly concerned with the approximation and analytical properties of individual offset curves, we are

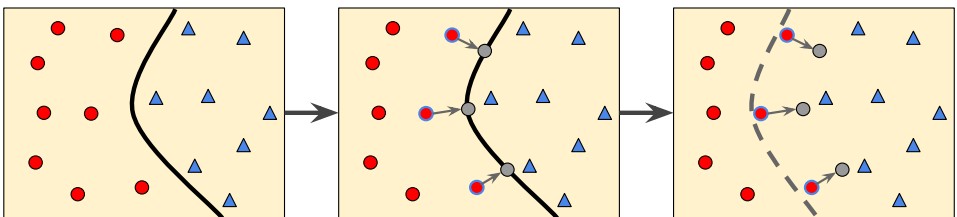

Figure 1: **(Left)** Original non-linear classifier $h$. **(Middle)** Strategic response to original classifier. **(Right)** Resulting *effective classifier* $h_\Delta$ (different from $h$) after accounting for strategic responses.

interested in the effects on class complexity and what this implies for learning. In terms of theory, both [33, 36] apply PAC analysis to strategic classification, the latter providing VC bounds for the linear class. Most relevant to us is [6], who study the general learnability of strategic learning under different feedback models. While not restricted to particular classes, their analysis considers graph costs, where users can move only to a finite set of possible points, and their results depend crucially on the cardinality of this set. Our work tackles the more pervasive setting of continuous cost functions, which imply continuous modifications. The above works focus primarily on statistical hardness; while this is also one of our aims, we are interested in establishing a broader understanding of the non-linear regime.

## 3   Preliminaries

We work in the original supervised setting of strategic classification [3, 16]. Let $x \in \mathcal{X} = \mathbb{R}^d$ denote user features, $y \in \mathcal{Y} = \{0, 1\}$ their corresponding label, and assume pairs $(x, y)$ are drawn iid from some unknown joint distribution $D$. Given a training set $T = \{(x_i, y_i)\}_{i=1}^m \sim D^m$, learning aims for the common goal of finding a classifier $h \in H$ that maximizes the expected predictive accuracy, where $H$ is some chosen model class. The unique aspect of strategic classification is that at test time, users modify their features to maximize gains in response to the learned $h$ via the *best-response mapping*

$$x^h := \Delta_h(x) = \operatorname*{argmax}_{x'} \alpha h(x') - c(x, x') \tag{1}$$

where $\alpha$ determines their utility from obtaining a positive prediction $\hat{y} = h(x)$, and $c$ is a function governing modification costs. We assume w.l.o.g. that the user gains no utility from a negative prediction. Like most works, we focus mainly on the $\ell_2$ cost, i.e., $c(x, x') = \|x - x'\|_2$, but discuss other costs in a subset of our results (see Appendix C.1 for discussion on how our results generalize to other cost functions). Because no rational user will pay more than $\alpha$ to obtain a positive prediction, $x^h$ will always be contained in the ball of radius $\alpha$ around $x$, denoted $B_\alpha(x) = \{x' : c(x, x') \leq \alpha\}$. It will also be useful to define the corresponding sphere, $S_\alpha(x) = \{x' : c(x, x') = \alpha\}$.

Given the above, the learning objective is to maximize the expected *strategic accuracy*, defined as:

$$\operatorname*{argmax}_{h \in H} \mathbb{E}_D[\mathbb{1}\{y = h(\Delta_h(x))\}] \tag{2}$$

**Levels of analysis.**   We assume w.l.o.g. that classifiers are of the form $h(x) = \mathbb{1}\{f(x) \geq 0\}$ for some scalar score function $f$, and that the decision boundary is then the surface of points for which $f(x) = 0$. Our analysis takes a 'bottom-up' approach in which we first consider changes to individual points on the boundary, then ask how such local changes aggregate to affect a given classifier $h$, and finally, how these changes impact the entire model class $H$. It will therefore be useful to consider three mappings between original and corresponding strategic objects. For points, the mapping is $z \mapsto \nabla_h(z) = \{x : \Delta_h(x) = z \wedge c(x, z) = \alpha\}$, i.e., the set of points $x$ that, in response to $h$, move to $z$ at maximal cost. For classifiers, we define the *effective classifier* as $h_\Delta(x) := h(\Delta_h(x))$ (see Fig. 1), and consider the mapping $h \mapsto h_\Delta$. Since both $h$ and $h_\Delta$ are defined w.r.t. raw (unmodified) inputs $x$, this allows us to compare the original decision boundary of $h$ to the 'strategic decision boundary' of its induced $h_\Delta$. For model classes, we consider the mapping $H \mapsto H_\Delta := \{h_\Delta : h \in H\}$, and refer to $H_\Delta$ as the *effective model class*.

**The linear case.**   As noted, most works in strategic classification consider $H$ to be the class of linear classifiers, $h(x) = \mathbb{1}\{w^\top x \geq b\}$. A known result is that $\Delta_h(x)$ can be expressed in closed form as a

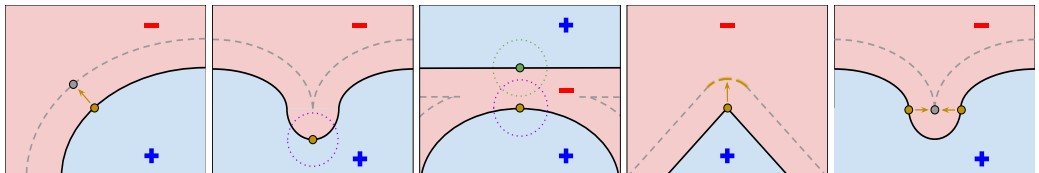

Figure 2: **Types of point mappings.** From left to right: *one-to-one* (case #1), *direct wipeout* (case #2), *indirect wipeout* (case #2), *expansion* (case #3), and *collision* (case #4). Black lines show original $h$, dashed gray lines indicate $h_\Delta$, gold arrows show $z \mapsto \nabla_h(z)$, and dotted circles show $S_\alpha(x)$

conditional projection operator: $x$ moves to $x_{\text{proj}} = x - \frac{w^\top x - b}{\|w\|_2^2} w$ if $h(x) = 0$ and $c(x, x_{\text{proj}}) \leq \alpha$, and otherwise remains at $x$ [9]. Note that all points that move do so in the same direction, i.e., orthogonally to $w$. Furthermore, each $x$ on the decision boundary of $h$ is mapped to a single, unique point $z$ on the decision boundary of $h_\Delta$ at an orthogonal distance (or cost) of exactly $\alpha$. Considering this for all boundary points, we get that $h_\Delta$ simply shifts the decision boundary by $\alpha$, which gives $h_\Delta(x) = \mathbb{1}\{w^\top x \geq b - \alpha\}$. Thus, $H_\Delta$ remains the set of all linear classifiers, and so $H_\Delta = H$.

**The non-linear case: first steps.** Informally stated, the main difference in the non-linear case is that points no longer move in the same direction (see Fig. 1). Note that while points still move to the 'closest' point on the decision boundary, the projection operator is no longer nicely defined. As a result, the shape of the decision boundary of $h$ may not be preserved when transitioning to $h_\Delta$. Moreover, as we will show, some points on the decision boundary of $h$ will 'disappear' and have no effect on the induced $h_\Delta$, while other points may be expanded to affect a continuum of points on $h_\Delta$. A final observation is that directionality is important in the strategic setting: because points always move towards the positive region, the shape of $h_\Delta$ for some $h$ may be very different than that of the same classifier but with "inverted" predictions, $h'(x) = 1 - h(x)$. This lack of symmetry means that we cannot take for granted that the induced $H_\Delta$ will include both $h_\Delta$ and $1 - h_\Delta$, even if $H$ does.

## 4 Underlying Mechanisms

Due to the added complexity of the non-linear setting, we first present some basic results and observations related to the underlying mechanisms of the non-linear setting in order to build the necessary intuition for our higher-level results. We begin by examining the point mapping $z \mapsto \nabla_h(z)$ and then turn to investigate how strategic behavior affects the classifier mapping $h \mapsto h_\Delta$.

### 4.1 Point mapping

To gain an understanding of what determines the shape of $h_\Delta$ of a general $h$, we begin by considering a single point $z$ on the decision boundary of $h$ and examining how (and if) it maps to corresponding point(s) on the decision boundary of $h_\Delta$ through the point mapping $z \mapsto \nabla_h(z)$. We identify and discuss four general cases, illustrated in Fig. 2:

**Case #1: One-to-one.** For linear $h$, the point mapping is always one-to-one. For non-linear $h$, the mapping remains one-to-one as long as $h$ is 'well-behaved' around $z$. We consider this the 'typical' case and note that each $z$ is mapped to its $\alpha$-offset point $x = z - \alpha \hat{n}_z$, where $\hat{n}_z$ is the unit normal at $z$ w.r.t. $h$ (see Appendix B.1). A necessary condition for *one-to-one* mapping is that $h$ is smooth at $z$, but smoothness is not sufficient, as the following cases depict.

**Case #2: Wipeout.** Even if $h$ is smooth at $z$, in some cases the point mapping will be null: $z \mapsto \varnothing$. We refer to this phenomenon as *wipeout*, and identify two types. *Direct wipeout* occurs when the signed curvature $\kappa$ at $z$ is negative and of sufficiently large magnitude; this is defined formally in Sec. 4.2. Intuitively, this results from strategic behavior obfuscating any rapid changes in the decision boundary of $h$. *Indirect wipeout* occurs when the set of points $\nabla_h(z)$ is contained in $B_\alpha(x')$ for some other $x'$ on the decision boundary. For both types, the result is that $B_\alpha(z)$ is entirely contained in the positive region of $h_\Delta$ and such $z$ have no effect on $h_\Delta$ (see Appendix B.2).

**Case #3: Expansion.** When the decision boundary of $h$ at $z$ is not smooth (i.e., a kink or corner), $z$ can be mapped to a continuum of points on the effective boundary of $h_\Delta$. This, however, occurs only when the decision boundary is locally convex towards the positive region. The set of mapped

points $\nabla_h(z)$ are those on $S_\alpha(z)$ which do not intersect with any other $B_\alpha(x')$ for any other $x'$ on the boundary of $h$. Thus, the shape of the induced 'artifact' is partial to a ball. Partial expansion may occur if some of the expansion artifact is discarded through indirect wipeout (Appendix C.2).

**Case #4: Collision.** A final case is when a set of points $\{z_1...z_n\}$ on the boundary of $h$ are all mapped to the same single point $x$ on $h_\Delta$, i.e., $\nabla_h(z) = \nabla_h(z') = x$ for all $z, z'$ in the set. Typically, this happens at the transitions between well-behaved decision boundary segments and wiped-out regions. Collision often results in non-smooth kinks on $h_\Delta$, even if all source points were smooth on $h$.

## 4.2 Curvature

Building up to the level of classifiers, it is first useful to examine what happens to boundary points $z$ in terms of how $h$ behaves around them. We make use of the notion of *signed curvature*, denoted $\kappa$, which quantifies how much the decision boundary deviates from being linear at a certain point $x$. For $d = 2$, curvature is formally defined as $\kappa = 1/r$, where $r$ is the radius of the circle that best approximates the curve at $x$. For $d > 2$, we use $\kappa^* = \sup(K)$, where $K$ includes all directional curvatures at $x$ (see Appendix A.1). The sign of $\kappa$ measures whether the decision boundary bends (i.e., is convex) towards the positive region ($\kappa > 0$), the negative region ($\kappa < 0$), or is locally linear ($\kappa = 0$).

Let $z$ be a typical point on the decision boundary of $h$ which satisfies the *one-to-one* mapping detailed in Sec. 4.1. We borrow a classic result from the field of offset curves [12] to calculate the *effective signed curvature* of $h_\Delta$ at $x$, denoted $\kappa_\Delta$, as a function of the curvature $\kappa$ of $h$ at $z$.

**Proposition 1.** *Let $z$ be a point on the decision boundary of $h$ with signed curvature $\kappa \geq -1/\alpha$. The effective curvature of the corresponding $x$ on the boundary of $h_\Delta$ is given by:*

$$\kappa_\Delta = \kappa/(1 + \alpha\kappa) \tag{3}$$

When $d > 2$, each $\kappa \in K$ is mapped as in Eq. 3 to form $K_\Delta = \{\kappa/(1 + \alpha\kappa) : \kappa \in K\}$, and $\kappa_\Delta^*$ follows. See Appendix B.3 for proof and an illustration of how this impacts the shape of $h_\Delta$. Prop. 1 entails several interesting properties. First, strategic behavior preserves the direction of curvature, i.e., $\kappa \cdot \kappa_\Delta \geq 0$ always. In particular, $\kappa = 0$ entails $\kappa_\Delta = 0$, which explains why linear functions remain linear. Second, the effective curvature $\kappa_\Delta$ is monotonically increasing in $\kappa$, concave, and sub-linear. This implies that for $\kappa > 0$ the effective positive curvature *decreases*, while for $\kappa < 0$ the effective negative curvature *increases*. Thus, strategic behavior acts as a *directional* smoothing operator, i.e., the boundaries' rate of change is expected to decrease but only in the positive direction.

Finally, in the positive direction, $\kappa_\Delta$ is bounded from above, approaching $1/\alpha$ as $\kappa \to \infty$. This gives:

**Observation 1.** *No effective classifier $h_\Delta$ can have positive curvature $\kappa_\Delta$ greater than $1/\alpha$.*

When $d > 2$, Obs. 1 holds for each curvature $\kappa \in K$ and can be summarized as $\kappa_\Delta^* \leq 1/\alpha$. Though Prop. 1 only implies Obs. 1 at points on $h_\Delta$ that are typical, we will show in Sec. 4.4 that it is also true at any point on $h_\Delta$. The above holds irrespective of the shape and curvature of the original $h$. In particular, non-smooth points on $h$ (i.e., with $\kappa \to \infty$), such as an edge or vertex at the intersection of halfspaces, become smooth on $h_\Delta$. Conversely, in the negative direction, $\kappa_\Delta$ approaches $-\infty$ as $\kappa \to 1/\alpha$. Thus, negative curvature can grow arbitrarily large, and at $\kappa = -1/\alpha$ maps to a non-smooth "kink" in $h_\Delta$. $\kappa_\Delta$ is no longer defined when $\kappa$ crosses the asymptote at $-1/\alpha$ because such points are wiped out and have no corresponding $x$ on the boundary of $h_\Delta$. This is the exact case of *indirect wipeout* in Sec. 4.1 (see Appendix B.2 for proof and Appendix C.10 for practical use in learning).

**Observation 2.** *Any point $z$ on the decision boundary of $h$ with curvature $\kappa < -1/\alpha$ has no influence on $h_\Delta$. In higher dimensions, this is true for any point $x$ with $\inf(K) < -1/\alpha$.*

## 4.3 Containment

Because the decision boundary is a level set of some (non-linear) score function $f$, contained regions are not only possible, but realistic. Consider a given classifier $h$ where the decision boundary forms some bounded connected negative region $C \subset \mathbb{R}^d$ where $h(x) = 0 \ \forall x \in C$. Strategic behavior dictates that negative areas shrink, and so only a subset of $C$ will still be classified as negative in $h_\Delta$. If $C$ is sufficiently small, then it will disappear completely leaving $h_\Delta(x) = 1 \ \forall x \in C$.

**Observation 3.** *Any negative region $C$ fully contained in some $B_\alpha(x')$ will become positive in $h_\Delta$.*

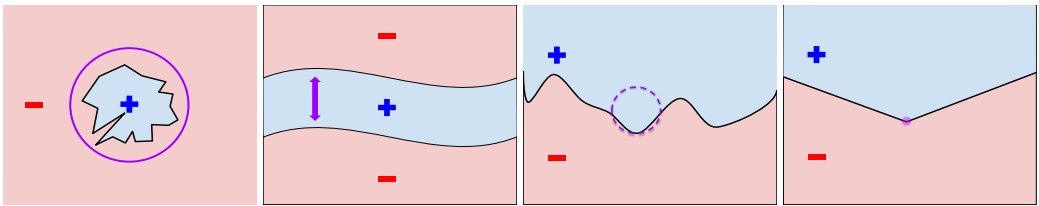

Figure 3: **Types of impossible effective classifiers.** From left to right: small positive region, narrow positive strip, large positive curvature, piecewise (hyper)linear convex towards the positive region.

The fact that positive areas grow and negative areas shrink also implies that there is a minimum positive region size that can exist in *any* $h_\triangle$. This idea will be formalized and generalized later in Sec. 4.4. One implication of Obs. 3 is that when negative regions are wiped out, two or more positive regions may merge, leading to a less expressive decision boundary. On the other hand, the strategic setting can also split a negative region, creating a more expressive decision boundary (see Appendix C.3).

**Observation 4.** *Strategic behavior can merge positive regions and split negative regions thereby either increasing or decreasing the number of connected regions (either positive or negative).*

### 4.4 Impossible Effective Classifiers

Based on the above, we derive two general statements about effective classifiers that cannot exist. Each statement considers some candidate classifier $g$, and shows that if $g$ satisfies a certain property, then there is no classifier $h$ for which $g$ is its induced effective classifier, i.e., $g \neq h_\triangle$ for any $h$. We first require some additional notation. Given $h$, define the set of negatively classified points as $\mathcal{X}_0(h) = \{x : h(x) = 0\}$. For a set $A \subseteq \mathcal{X}$, we say a point $x$ is *reachable* from $A$ if $\exists x' \in A : c(x, x') \leq \alpha$.

**Proposition 2.** *Let $g$ be a candidate classifier. If there exists a point $x$ on the decision boundary of $g$ such that all points $x' \in S_\alpha(x)$ are reachable from some point in $\mathcal{X}_0(g)$, then there is no $h$ such that $g = h_\triangle$.*

The intuition behind Prop. 2 is that no $h$ can simultaneously strategically classify $x$ as positive while all $y \in \mathcal{X}_0(g)$ as negative. Full proof in Appendix B.4. As stated in Prop. 3, when $g$ is smooth at $x$, Prop. 2 can be simplified to require that only a single point be checked instead of all of $S_\alpha(x)$.

**Proposition 3.** *Let $g$ be a candidate classifier. If there exists a point $x$ on the decision boundary of $g$ where (i) $g$ is smooth, and (ii) the offset point $\hat{x} = x + \alpha \hat{n}_x$ is reachable by some other $x' \in \mathcal{X}_0(g)$, then there is no $h$ such that $g = h_\triangle$.*

The intuition behind Prop. 3 is the same as for Prop. 2, except that when $g$ is smooth at $x$, the only point in $S_\alpha(x)$ that may not be reachable by some $x' \in \mathcal{X}_0(g)$ is the $\alpha$-offset (see Appendix B.5).

**Examples.** The implication of Props. 2 and 3 is that there exist decision boundaries that are generally realizable, but are no longer realizable as effective classifiers in the strategic setting. Note that because it only takes a single decision boundary point to render an entire effective classifier infeasible, each of these examples are broad categories and pose meaningful restrictions on the set of possible effective classifiers. As illustrated in Fig. 3, we highlight four types of impossible effective decision boundaries:

1. Classifiers with a small positive region that can be enclosed by a ball of radius $\alpha$.
2. Classifiers with a narrow positive region in which points are $\alpha$-close to the decision boundary.
3. Classifiers with large positive signed curvature anywhere on the decision boundary.
4. Classifiers with any locally convex piecewise linear or hyperlinear segments.

All proofs are in Appendix C.4, and either build on Obs. 1-4, or show the conditions of Props. 2 or 3 hold. From these four examples, we note the following observation.

**Observation 5.** *The mapping $h \mapsto h_\triangle$ is not bijective, with many candidate $h_\triangle$ having no corresponding $h$ and many $h$ mapped to the same $h_\triangle$.*

In Appendix C.5 we show that there are actually infinite $h$ that map to nearly all possible $h_\triangle$. Furthermore, Obs. 5 will be important for proving Thm. 1 and Cor. 1.

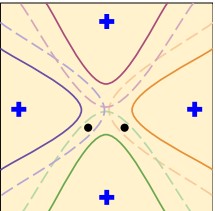 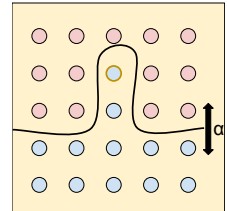

Figure 4: **(Left)** Example of increasing VC. **(Right)** Example of a dataset with limited strategic accuracy. Any $h_\Delta$ which correctly classifies the gold-rimmed point must err on a negative point.

## 5 Class-level analysis

On the classifier level, the strategic setting is clearly more restrictive than the regular setting. Indeed, in Sec. 5.2 we extend this idea into the model class level by analyzing the impacts on universality and optimal accuracy. However, as we show in Sec. 5.1, the model class mapping is more nuanced than the classifier mapping since complexity can actually increase as a result of strategic behavior.

### 5.1 Class complexity

Recall that for linear classes, $H_\Delta$ remains linear, and so $H \mapsto H_\Delta = H$. Although this equivalence is not unique to the class of linear classifiers (see examples in Appendix C.6), typically, we can expect strategic behavior to alter the complexity of the induced class. Here, we use VC analysis to shed light on this effect by examining the possible relations between $\mathrm{VC}(H)$ and $\mathrm{VC}(H_\Delta)$.[1]

**Complexity reduction.** Since our results so far point out that effective classifiers are inherently constrained, a plausible guess is that effective complexity should decrease. Indeed, there are some natural classes where the VC cannot increase. For example:

**Proposition 4.** *Let $Q_s^k$ be the class of negative-inside $k$-vertex polytopes bounded by a radius of $s$, then $\mathrm{VC}(Q_s^k) \geq \mathrm{VC}(Q_{s,\Delta}^k)$. Moreover, if $s = \infty$, then $\mathrm{VC}(Q_s^k) = \mathrm{VC}(Q_{s,\Delta}^k)$ as implied by Thm. 2.*

The proof for Prop. 4 can be found in Appendix B.7. Curiously, in some extreme cases, the VC reduction spans the maximum extent.

**Theorem 1.** *There exist several model classes $H$ where $\mathrm{VC}(H) = \infty$, but $\mathrm{VC}(H_\Delta) = 0$.*

The proof for Thm. 1 can be found in Appendix B.8. See Appendix C.7 for other examples where VC strictly decreases but does not diminish completely. Thm. 1 carries concrete implications for strategic learning:

**Corollary 1.** *Some model classes that are not learnable become learnable in strategic environments.*

**Complexity increase.** Though a lower effective VC is indeed a possibility, a more plausible outcome is for it to stay the same or increase. As an intuition for how the VC might increase, consider the simple example in Fig. 4 (left). Here, $H$ includes four non-overlapping classifiers (each color is a separate classifier) with $\mathrm{VC}(H) = 1$. Yet, with strategic behavior, the effective classifiers (dashed curves) overlap, increasing the shattering capacity to 2. This case can be generalized to give $VC(H_\Delta) = \theta(d \cdot VC(H))$ (see Appendix C.8). We next identify a simple generic sufficient condition for the effective complexity of any learnable class to be non-decreasing.

**Theorem 2.** *If $H$ is closed under input scaling and $\mathrm{VC}(H) < \infty$, then $\mathrm{VC}(H) \leq \mathrm{VC}(H_\Delta)$.*

The proof for Thm. 2 can be found in Appendix. B.9. Thm. 2 applies to many common classes, including polynomials, piecewise linear functions, and most neural networks. See Appx. C.10 for implications on model selection.

Because Thm. 2 suggests that the effective VC can increase, but does not state by how much, one concern would be that this increase can be unbounded, which would deem the effective class unlearnable. Our next result shows that this is not the case even for the common and highly expressive class of piecewise-linear classifiers (with a bounded number of segments), which includes neural

---

[1]This aligns with the notion of *strategic VC* used in [22, 33], i.e., $\mathrm{SVC}(H) \equiv \mathrm{VC}(H_\Delta)$.

networks with ReLU activations (of fixed size) as a special case, and is often used in studies on neural network approximation [e.g., 19, 28].

**Theorem 3.** *Let $H^{m,k}$ be a class of piecewise-linear classifiers with at most m segments and k intersections of linear segments. Then $\mathrm{VC}(H_\Delta^{m,k}) = O\big(d\,m\,log(m) + \nu_p\,k\,log(k)\big)$, where $\nu_p$ is the VC of the class of $\ell_p$ norm balls, and is $\Theta(d)$ for $p = 1, 2, \infty$.*

The proof for Thm. 3 is in Appendix B.10. We also prove that $\mathrm{VC}(H_{m,k}) = O\big(d\,m\,log(m)\big)$, suggesting that the effective VC dimension can increase, but only to a limited extent. The main implication of Thm. 3 is therefore that under these conditions, strategic behavior maintains learnability: for any $H$ that can be expressed as (or approximated by) piecewise-linear functions, if $H$ (or its approximation) is learnable, then $H_\Delta$ is also learnable. We conjecture that this relation remains true for general $H$; see Appendix A.3 for discussion on the challenges of the general case and connections to [6]. For the special case of positive-inside polytopes — a subset of the piecewise-linear class — the effective VC dimension even maintains the same order of magnitude.

**Theorem 4.** *Consider either the $\ell_1$, $\ell_2$, or $\ell_\infty$ costs and let $H \subseteq P^k$, where $P^k$ is the set of all k-vertex polytopes in $R^d$ and has $\mathrm{VC}(P^k) = O(d^2\,k\,\log k)$[23]. Then $\mathrm{VC}(H_\Delta) = O(d^2\,k\,\log k)$.*

The proof for Thm. 4 is in Appendix. B.11. Note that for the $\ell_1$ and $\ell_\infty$ costs, the effective class remains within the polytope family, but potentially with additional vertices (see Appendix B.6). For the $\ell_2$ cost, the bound still holds despite $H_\Delta$ no longer including only polytopes (since corners in $h$ can become "rounded" in $h_\Delta$).

## 5.2 Universality and approximation

Our results in Sec. 4.4 show four 'types' of decision boundaries that cannot be attributed to any effective classifier, though others may exist. This implies that several basic classifiers are not realizable in the strategic setting and cannot be approximated well. This drives our principal conclusion:

**Corollary 2.** *Any universal approximator class $H$ is no longer universal under strategic behavior.*

In other words, there exists a classifier $g$ for which no effective class $H_\Delta$ can include $g$ as a member, even for $H$ that are universal approximators in the non-strategic setting. Common examples of universal approximator classes include polynomial thresholds [32], RBF kernel machines [27], many classes of neural networks [e.g., 17], and gradient boosting machines [13]. In the strategic setting, these classes are no longer all-encompassing like in the standard setting.

**Approximation gaps.** Cor. 2 implies that even if the data is in itself realizable, and even without any limitations on the hypothesis class, the learner may already start off with a bounded maximum training accuracy simply by learning in the strategic setting. Fig. 4 (right) depicts one such example dataset where the maximum attainable accuracy drops from 1 to 0.96. This is because strategic behavior makes it impossible to generally approximate intricate functions, e.g., by using fast-changing curvature (via increased depth) or many piecewise-linear segments (with ReLUs). Another result is that the common practice of interpolating the data using highly expressive over-parametrized models (e.g., as in deep learning) is no longer a viable approach. On the upside, it may be possible for the strategic responses to prevent unintentional overfitting, allowing for better generalization.

In extreme cases, the gap between the maximum standard accuracy and the maximum strategic accuracy can be quite large. Our next result demonstrates that there are distributions in which, despite a maximum standard accuracy of 1, the maximum strategic accuracy falls to the majority class rate.

**Proposition 5.** *For all d, there exist (non-degenerate) distributions on $\mathbb{R}^d$ that are realizable in the standard setting, but whose maximum strategic accuracy is the majority class rate, $\max_y p(y)$.*

In one dimension, the distribution can be constructed by placing a negative point $\delta$ to the left and right of each positive point. In higher dimensions, negative points can be arranged in an $\mathbb{R}^d$ simplex around each positive point. The full proof is in Appendix B.12. Prop. 6 extends this idea to show that there is a distribution whose maximum strategic accuracy approaches $1/2$ as $\alpha$ increases. The general proof is given in Appendix B.13.

**Proposition 6.** *There exists a distribution that is realizable in the standard setting, but whose accuracy under any $h_\Delta$ is at most $0.5 + \frac{1}{\lfloor 2\alpha+1 \rfloor}$.*

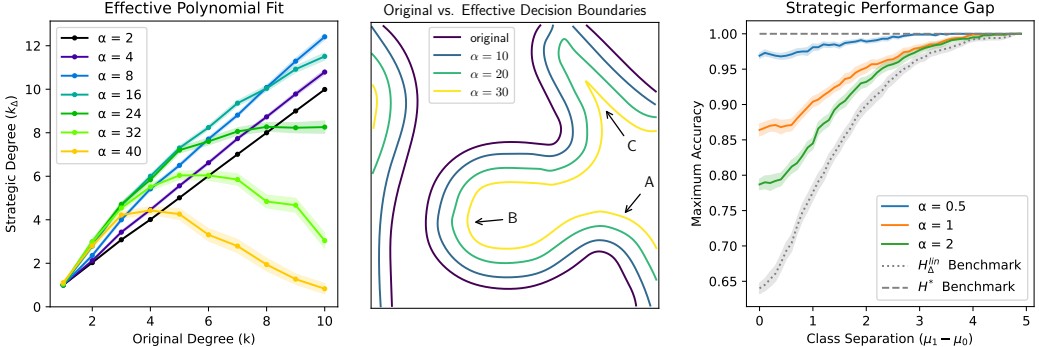

Figure 5: **(Left) Expressivity.** For random polynomial classifiers of degree $k$, results show the smallest degree $k'$ that captures the effective decision boundary. **(Center)** An instance showing positive curvature decreasing (A), negative curvature increasing (B), and wipeout (C). **(Right) Approximation.** As data becomes more entangled (low separation), strategic approximation degrades.

In practice, we expect the accuracy gap to depend on the distribution. We explore this gap experimentally in Sec. 6. An interesting observation is that in certain non-realizable settings, the maximum strategic accuracy can *exceed* the maximum standard accuracy. (see Appendix C.9).

# 6 Experiments

To complement our theoretical results, we present two experiments that empirically demonstrate the effects of strategic behavior on non-linear classifiers. The code for both experiments is available at https://github.com/BML-Technion/scnonlin.

## 6.1 Expressivity

Following our results from Sec. 5.1 on the change in hypothesis class expressivity, we experimentally test whether the $h_\Delta$ of a given random $h$ belongs to a more or less expressive class than $h$ itself. In particular, we focus on degree-$k$ polynomial classifiers and compare the degree $k$ of $h$ to the degree $k_\Delta$ of the polynomial approximation of $h_\Delta$.[2] Because user features are often naturally bounded, we assume that $x \in \mathcal{X} = R^2$ and that $x$ is bounded by $|x|_\infty \leq 100$. See Appendix D.1 for full details.

Fig. 5 (left) shows results for varying $k \in [1, 10]$ and for $\alpha \in [2, 40]$. When $\alpha$ is small, we find that $k_\Delta \approx k$, meaning that strategic behavior has little impact on expressivity. However, as $\alpha$ increases, $k_\Delta$ becomes larger than $k$, suggesting that $h_\Delta$ is more complex than $h$. This is due to the fact that as $\alpha$ increases, point mapping collisions (see Sec. 4.1) become more likely, causing non-smooth cusps — which are more complex than basic polynomials — to form in the decision boundary (Fig. 5 (center)). When $\alpha$ is quite large, we see that $k_\Delta$ is still larger for low $k$, but drops considerably for higher $k$. This can be attributed to the fact that tightly-embedded higher-dimension polynomials and increased strategic reach from large $\alpha s$ are both causes of *indirect wipeout* (see Sec. 4.1) because they increase the reachability of $\nabla_h(z)$ by other decision boundary points. As such, much of the original decision boundary is wiped out, leaving behind a lower complexity effective decision boundary.

## 6.2 Approximation

As seen in Sec. 5.2, the lack of universality in the strategic setting can impose a limitation on the maximal attainable strategic accuracy. To demonstrate the practical implications of this effect, we upper bound the maximum strategic accuracy of any $H_\Delta$ on a set of synthetic data by experimentally calculating the maximum strategic accuracy of the unrestricted effective hypothesis class $H_\Delta^*$. We compare the results to two benchmarks: (i) the standard accuracy of the regular unrestricted hypothesis class $H^*$ (which is always 1), and (ii) the strategic (and standard) accuracy of the linear effective hypothesis class $H_\Delta^{\text{lin}} = H^{\text{lin}}$. The first benchmark measures the extent to which the strategic setting

---

[2]Though the $h_\Delta$ of a polynomial $h$ is not necessarily a polynomial, we can compare the expressivity of $h$ and $h_\Delta$ by finding the lowest degree-$k_\Delta$ polynomial $g$ that well-approximates $h_\Delta$.

hinders the learner, while the second measures the extent to which the learner can benefit from non-linearity, even in the strategic world. We generate data by sampling points from class-conditional Gaussians $x \sim \mathcal{N}(y\mu, 1)$, and aim to find an $h$ that obtains an optimal fit. The parameter $\mu$ serves to show how $H_\Delta$ compares to our two baselines under data that ranges from well-separated (large $\mu$) to more "interleaved" and harder to classify by restricted classes (small $\mu$). Details in Appendix D.2.

Fig. 5 (right) shows results for increasing class separation ($\mu$). As $\mu$ decreases and the classification problem becomes more difficult, the maximum strategic accuracy of $H_\Delta^*$ diverges from the upper $H^*$ baseline, with performance deteriorating as strategic behavior intensifies (i.e., larger $\alpha$). Because any actual $H_\Delta$ can only have worse performance than the ideal $H_\Delta^*$, this indicates that strategic behavior can become a significant burden in more difficult tasks, such as classification in a non linearly-separable setting. Nonetheless, when $\mu$ is small, $H_\Delta^*$ significantly outperforms the lower $H_\Delta^{\mathrm{lin}}$ baseline, signaling that non-linearities still improve accuracy in the strategic setting despite any strategic effects.

## 7   Discussion

This work sets out to explore the interplay between non-linear classifiers and strategic user behavior. Our analysis demonstrates that non-linearity induces behavior that is both qualitatively different than the linear case and also non-apparent by simply studying it. Simply put, the strategic setting is fundamentally limited, though has a few potential advantages. Our results show how such behavior can impact classifiers by morphing the decision boundary and model classes by increasing or decreasing complexity. Although prior work has made clear the importance of accounting for strategic behavior in the learning objective, our work suggests that there is a need for additional broader considerations throughout the entire learning pipeline from the choices we make initially (such as which model class to use) to setting our final expectations (such as the accuracy levels we may aspire to achieve). We detail practical takeaways for learning and model class selection in Appendix C.10. This motivates future theoretical questions, as well as complementary work on practical aspects such as optimization (i.e., how to solve the learning problem), modeling (i.e., how to capture true human responses), and evaluation (i.e., on real humans and in the wild).

**Acknowledgments**

The authors are grateful to Shay Moran and Nadav Dym for thoughtful discussions and suggestions. This work is supported by the Israel Science Foundation grant no. 278/22.

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

# A  Background

## A.1  Signed Curvature

Our analysis makes use of the notion of *signed curvature* — a signed version of the normal curvature — to quantify shape alteration. Because there is an inherent notion of direction due to the binary nature of class labels, we can define curvature as being positive if the curvature normal vector points in the direction of the positive region and negative if it points in the direction of the negative region. Though in $\mathbb{R}^2$ this defines a single value, we must instead define a curvature value for each 'direction' in higher dimensions. More formally, for each tangent vector $\vec{t}$ in the tangent hyperplane $T_x$ at $x$, we define the curvature value $\kappa_{\vec{t}}$ as the signed curvature at $x$ on the curve obtained by intersecting the decision boundary with the plane $L$ containing $\vec{t}$ and the normal vector $\hat{n}_x$. Defining $K = \{\kappa_{\vec{t}} : \vec{t} \in T\}$ as the set of all such curvatures, our analysis will focus on $\kappa^* = \sup(K)$ in $\mathbb{R}^3$ and above.

## A.2  Bounded/Unbounded Input Domain

At a high level, our results in Sec. 5 do not require an unbounded input space, but rather some input space that is larger than $\mathcal{X}$. In particular, if we assume that raw inputs, $x$, are bounded by $|x| \leq B$, then for the problem to generally be well-defined, we must allow this space to expand at the minimum to $B + \alpha$ to accommodate for strategic input modifications. More generally, we must ensure that $|\Delta_h(x)| \leq B(\alpha)$ holds for all $x$ and any $h$, for some expansion $B(\alpha) \geq B$, where the exact form of $B(\alpha)$ depends both on the cost function and on $H$.

Other than Thm. 2, our results simply require the minimal expansion be satisfied and are therefore not dependent on an unbounded input domain. Likewise, Thm. 2 does not require an unbounded input domain since for any given learnable $H$, there exists some finite $B(\alpha)$ for which the result holds. However, since the statement is generic (i.e., applies to all learnable $H$ that are closed to scaling), we do not have a closed-form expression for $B(\alpha)$ that applies simultaneously to all such $H$.

## A.3  General Upper Bound on VC Increase

In Thm. 3, we provide an upper bound on the strategic VC dimension of classes of piecewise-linear classifiers, though note that a general upper bound remains an open question. A key difficulty in finding a general upper bound in the strategic setting is that points will move differently when facing different $h$ from the same $H$. Thus, reasoning about the shattering of an arbitrary set must account for how points can move under all possible $h \in H$, though not simultaneously, which proved to be highly challenging in the general, unstructured case. This is why we have included results that exploit common structures like the upper bound based on piecewise linearity (Thm. 3) and lower bound based on $H$ that are closed to scaling (Thm. 2).

That said, we conjecture that Thm. 3 holds for general $H$, as proven by Cohen et al. [6] in the special case of finite manipulation graphs. In particular, they show that $\mathrm{VC}(H_\Delta) = \tilde{\Theta}(\mathrm{VC}(H) \cdot log\, k)$ where $k$ is the degree of the manipulation graph. Note that the bound is not transferable to our case since it is vacuous for $k \to \infty$, and the proof technique does not apply to continuous graphs.

# B  Proofs

**Lemma 1.** *In $\mathbb{R}^2$, if the curvature $\kappa$ at $x$ is positive, then the radius $r_{insc}$ of the maximum inscribed circle at $x$ is no bigger than the radius $r_{osc} = \frac{1}{\kappa}$ of the osculating circle at $x$ (the circle that best approximates the curve at $x$ and has the same curvature). As a corollary, if curvature $\kappa$ at $x$ is negative, then the radius $r_{esc}$ of the maximum escribed circle at $x$ is no bigger than the radius $r_{osc} = -\frac{1}{\kappa}$ of the osculating circle at $x$.*

*Proof.*  Note that both the osculating circle and the inscribed circle have their centers along the normal line to the curve at $x$ (and on the same side of $h$), and that the osculating circle may cross the curve at $x$. If it does, then any circle which has a larger radius and a center on the same side of $h$ along the normal line to the curve will also cross the decision boundary at $x$, so the radius of the inscribed circle (which strictly does not cross the curve) is strictly smaller than the radius of the osculating circle. If the osculating circle does not cross the curve, then it must be the largest strictly tangent

circle at $x$, or else it would not be the best circular approximation to the curve at $x$. As such, the largest inscribed circle, which must be both tangent to the curve at $x$ also not intersect it anywhere else, must have a radius that is no larger than $r_{osc}$.

$\square$

## B.1 One-to-One Mapping

**Claim:**  *In the one-to-one mapping, the point $z$ on the decision boundary of $h$ is mapped to the point $x = z - \alpha \hat{n}_z$ on the decision boundary of $h_\Delta$.*

*Proof.* Note that all points on the decision boundary of $h_\Delta$ must be at the maximum cost away from $h$[3], so $\nabla_h(z) \subset S_\alpha(z)$. Additionally, for each $x \in \nabla_h(z)$, the ball $B_\alpha(x)$, which is the set of points that $x$ can move to, must not intersect the decision boundary of $h$ (or else it would not be at maximum cost away) and must furthermore be strictly tangent to the decision boundary of $h$ at $z$. Given that the decision boundary of $h$ is smooth at $z$, this means that the normal to the curve at $z$ must run along the radius of $B_\alpha(x)$ from $z$ to $x$, implying that $x = z \pm \alpha \hat{n}_z$. Because of the inherent directionality of the setting, the normal vector at $z$ is defined to point in the positive direction. This means that only the point $x = z - \alpha \hat{n}_z$ would seek to move to $z$ to gain a positive classification, as $x = z + \alpha \hat{n}_z$ is either already labeled positive or can move to a closer point. Therefore, $\nabla_h(z) = \{z - \alpha \hat{n}_z\}$ in the one-to-one case. $\square$

## B.2 Wipeout Mapping

**Claim:**  *Points on the decision boundary of $h$ that are 'wiped out' do not affect the effective decision boundary of $h_\Delta$.*

*Proof.* In the case of *indirect wipeout*, this follows directly from the definition: if $\nabla_h(z)$ is contained in $B_\alpha(x')$ for some other $x'$ on the decision boundary of $h$ (or in the union of several $B_\alpha(x')$), then none of the points in $\nabla_h(z)$ are at a maximum cost away from $h$, and are therefore not on the decision boundary of $h_\Delta$.

In the case of *direct wipeout*, the normal *one-to-one* mapping fails because the point $x = z - \alpha \hat{n}_z$ is no longer at the maximum cost away from $h$. Specifically, because the curvature $\kappa < -1/\alpha$ [4], the ball $B_\alpha(x)$ now intersects $h$, which means that $x$ is less than $\alpha$ away from $h$ and therefore not on the decision boundary of $h_\Delta$. For $\mathbb{R}^2$, Lemma 1 proves that the radius of the maximum escribed circle at $z$ is less than the radius of the osculating circle, so $r_{esc} \leq r_{osc} = -\frac{1}{\kappa} < \alpha$. Therefore, there is no circle of radius $\alpha$ that is strictly tangent to $h$ (on the negative side), so $B_\alpha(x)$ must intersect $h$.

For higher dimensions (see Appendix A.1), there is no longer a single normal curvature. Instead, we define the set $K$ of curvatures in each plane $p \in P$ defined by the normal vector and some tangent vector at $z$. Note that $B_\alpha(x)$ will intersect $h$ if any of its projections onto each $p \in P$ (a circle) intersects the curve obtained by intersecting the decision boundary with the plane $p$. Since each of these cases are in $\mathbb{R}^2$, $B_\alpha(x)$ will intersect $h$ if $\inf(K) < -1/\alpha$.

$\square$

## B.3 Proof of Proposition 1

**Proposition 1:**  *Let $z$ be a point on the decision boundary of $h$ with signed curvature $\kappa \geq -1/\alpha$. The effective curvature of the corresponding $x$ on the boundary of $h_\Delta$ is given by:*

$$\kappa_\Delta = \kappa/(1 + \alpha\kappa) \tag{3}$$

*Proof.* While an equivalent form of Prop. 1 has already been proven in the field offset curves [12], we nonetheless reprove it here to offer some intuition behind it. In the normal, *one-to-one* mapping, because both the center of curvature and $\nabla_h(z)$ lie on the normal line to $h$ at $z$ (see Appendix B.1),

---

[3]Otherwise, $x$ would not be on the boundary of positive and negative classification by $h_\Delta$ because all of the points adjacent to it can also strategically move to achieve a positive classification.

[4]or $\inf(K) < -1/\alpha$ for $d > 2$. See Sec. 4.2 and Appendix A.1 for curvature definitions

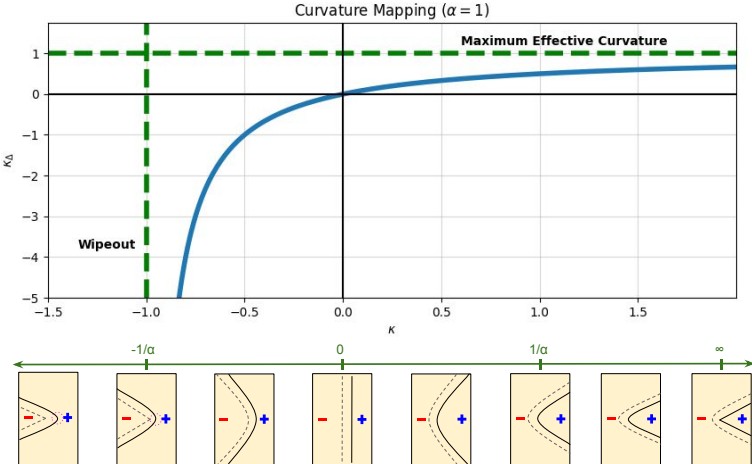

Figure 6: **(Top)** Graph of Eq. 3 with asymptotes represented as dashed lines. **(Bottom)** Depictions of curvature mappings for increasing $\kappa$.

strategic movement can be seen as increasing the signed radius of curvature, $r_{osc}$, by $\alpha$ (where we will define the radius of curvature as negative if the curvature is negative at $z$). In higher dimensions, this is equivalent to increasing each of the radii of curvature by $\alpha$ (see Appendix A.1). We therefore have:

$$r_{osc,\Delta} = r_{osc} + \alpha$$
$$\kappa_\Delta = \frac{1}{r_{osc,\Delta}} = \frac{1}{r_{osc} + \alpha}$$
$$= \frac{1}{\frac{1}{\kappa} + \alpha}$$
$$= \frac{\kappa}{1 + \alpha\kappa}$$

Fig. 6 shows the graph of Eq. 3 as well as examples of curvature mappings for various $\kappa$. Note that Eq. 3 has a horizontal asymptote at $\kappa_\Delta = 1/\alpha$ because the effective curvature cannot exceed $1/\alpha$ (see Sec. 4.4). Additionally, Eq. 3 has a vertical asymptote at $\kappa = -1/\alpha$ because any point with lower curvature has no effect on $h_\Delta$ (see Appendix B.2). When $\kappa = -1/\alpha$, $x$ is mapped to a non-smooth kink of infinite curvature. $\square$

### B.4  Proof of Proposition 2

**Proposition 2:** *Let $g$ be a candidate classifier. If there exists a point $x$ on the decision boundary of $g$ such that all points $x' \in S_\alpha(x)$ are reachable from some point in $\mathcal{X}_0(g)$, then there is no $h$ such that $g = h_\Delta$.*

*Proof.* In order for $g$ to be the effective classifier of the regular classifier $h$, we must have:

1. For each point $q$ that is positively classified by $g$, there must be a point $q' \in B_\alpha(q)$ which is classified as positive by $h$. In other words, $q$ must be able to acquire a positive label by staying in place or strategically moving within $B_\alpha(q)$.

2. For each point $q$ that is negatively classified by $g$, there cannot be any points $q' \in B_\alpha(q)$ which are classified as positive by $h$. In other words, $q$ must not be able to acquire a positive label by staying in place or strategically moving within $B_\alpha(q)$.

Assume there exists a point $x$ on the decision boundary of $g$ such that all points $x' \in S_\alpha(x)$ are reachable from $\mathcal{X}_0(g)$ and that there exists $h$ such that $g = h_\Delta$. Since $x$ is on the decision boundary

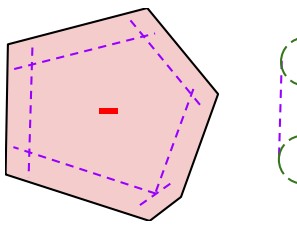
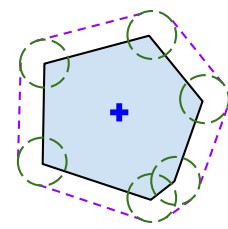

Figure 7: **(Left)** Each face of a negative-inside polytope is mapped to a new linear face, though the resulting $h_\triangle$ may be a polytope with fewer faces and vertices due to wipeout. **(Right)** Each face of a positive-inside polytope is also mapped to a new linear face, though each vertex will have to expand to fill the gap between these faces.

of $g$, there exists a point $x + \delta, \delta \to 0$ that is negatively classified by $g$. For requirements 1 and 2 to hold, there must be a point $x' \in S = \{q : q \in B_\alpha(x), q \notin B_\alpha(x + \delta)\}$ that is positively classified by $h$. Since $S \subset S_\alpha(x)$, we can expand the claim to say that there must be a point $x' \in S_\alpha(x)$ that is positively classified by $h$. However, if all points $x' \in S_\alpha(x)$ are reachable by some point $q$ that is negatively classified by $g$, then none of the points $x' \in S_\alpha(x)$ can be positively classified by $h$ (by requirement 2). Therefore, if there exists a point $x$ on the decision boundary of $g$ for which every point on the sphere $S_\alpha(x)$ is reachable by some point that is negatively classified by $g$, then $g$ cannot be a possible effective classifier. $\qquad\square$

### B.5 Proof of Proposition 3

**Proposition 3:** *Let $g$ be a candidate classifier. If there exists a point $x$ on the decision boundary of $g$ where (i) $g$ is smooth, and (ii) the offset point $\hat{x} = x + \alpha \hat{n}_x$ is reachable by some other $x' \in \mathcal{X}_0(g)$, then there is no $h$ such that $g = h_\triangle$.*

*Proof.* By Prop. 2, for each point $x$ on the decision boundary of $g$, there must be a point $x'$ on $S_\alpha(x)$ that is classified positively by $h$ and also not reachable by any point negatively classified by $g$. As a result, the ball $B_\alpha(x')$, which is the set of points that can move to $x'$, must not intersect the decision boundary and must furthermore be strictly tangent to the decision boundary at $x$. Given that the decision boundary is smooth at $x$, this means that the normal to the curve at $x$ must run along the radius of $B_\alpha(x')$ from $x$ to $x'$, implying that $x' = x \pm \alpha \hat{n}_x$. Because the strategic movement is in the direction of a positive classification, the only possible option is for $x$ to move to $x' = x + \alpha \hat{n}_x$. However, if $x' = x + \alpha \hat{n}_x$ is reachable by some point that is negatively classified by $h_\triangle$, then the associated $h$ would not have classified $x'$ as positive, so $x$ would not have been a positively classified point by $h_\triangle$. $\qquad\square$

### B.6 Effective Class of Polytope Family

Each polytope is simply a combination of $n$ linear faces. As such, when a polytope has a negative inside, each face will be mapped to a new linear face (with some wipeout of part or all of the face) leading to an $h_\triangle$ that is a smaller-radius polytope, albeit with potentially fewer faces and edges (see Fig. 7 (Left)). Although the faces of a positive-inside polytope also map to new linear faces, each vertex will have to expand to fill the gap between these new faces. As such, $h_\triangle$ is now a larger polytope whose corners are partial to a ball (see Fig. 7 (Right)). For the $\ell_1$ and $\ell_\infty$ costs, $h_\triangle$ therefore remains a polytope since the $\ell_1$ and $\ell_\infty$ balls are piecewise-linear themselves. For the $\ell_2$ cost, $h_\triangle$ is now a polytope with rounded corners. In either case, the decision boundary of each $p_\triangle^k \in P_\triangle^k$ can be seen as the intersection of a radius-scaled $p^k$ and $k$ balls centered at each vertex of $p^k$.

### B.7 Proof of Proposition 4

**Proposition 4:** *Let $Q_s^k$ be the class of negative-inside $k$-vertex polytopes bounded by a radius of $s$, then $\mathrm{VC}(Q_s^k) \geq \mathrm{VC}(Q_{s,\triangle}^k)$. Moreover, if $s = \infty$, then $\mathrm{VC}(Q_s^k) = \mathrm{VC}(Q_{s,\triangle}^k)$ as implied by Thm. 2.*

*Proof.* As noted in Appendix B.6, negative-inside polytopes map to smaller-radius negative-inside polytopes with potentially fewer faces and vertices. As such $Q_{s,\triangle}^k \subset Q_s^k$, so $\mathrm{VC}(Q_{s,\triangle}^k) \leq \mathrm{VC}(Q_s^k)$.

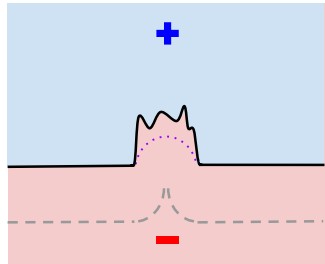

Figure 8: Example $h$ from a class $H$ where $\text{VC}(H) = \infty$, but $\text{VC}(H_\Delta) = 0$. The gray dashed curve is the $h_\Delta$ of any $h$ like the one above. Each such $h$ has the same small gap where the decision boundary remains above the dotted purple curve.

However, if $s = \infty$, the class $Q_s^k$ is closed to input scaling leading to $\text{VC}(Q_{s,\Delta}^k) \geq \text{VC}(Q_s^k)$ by Thm. 2, so we must have that $\text{VC}(Q_{s,\Delta}^k) = \text{VC}(Q_s^k)$. □

### B.8 Proof of Theorem 1

**Theorem 1:** *There exist several model classes $H$ where $\text{VC}(H) = \infty$, but $\text{VC}(H_\Delta) = 0$.*

*Proof.* We provide two examples. For the first example, fix $\alpha < 1$ and consider the class of $\mathbb{R}^2$ classifiers $H = \{h_S : S \subseteq \mathbb{Z}^2\}$ where $h_S = \mathbb{1}\{x \notin B_{\alpha-\delta}(s) \ \forall s \in S\}$. Each classifier in this model class contains small, non-overlapping negative regions centered at lattice points. Thus, the negative regions of all $h \in H$ will disappear under strategic movement, leaving $h_\Delta(x) = 1 \ \forall h \in H$. Despite the fact that $H$ shatters $\mathbb{Z}^2$ (and therefore has $\text{VC}(H) = \infty$), $H_\Delta$ cannot shatter a single point (and therefore has $\text{VC}(H_\Delta) = 0$).

Another example where $h_\Delta$ is perhaps more natural can be seen in Fig. 8. In this case, $H$ is the class of classifiers with narrow gaps whose decision boundaries can be written as

$$h_i\big((x_1, x_2)\big) = \begin{cases} 1 & |x_1| \geq k, \quad x_2 \geq 0 \\ 1 & |x_1| < k, \quad x_2 \geq f_i(x_1) \\ 0 & else \end{cases} \tag{4}$$

where $k$ is some constant less than $\alpha$ and each function $f_i(x_1)$ satisfies $f_i(x_1) \geq \sqrt{\alpha^2 - x_1^2} + \sqrt{\alpha^2 - k^2}$ for all $-k \leq x_1 \leq k$ (i.e., $f_i(x)$ lies above the radius-$\alpha$ arc between $(-k, 0)$ and $(k, 0)$). In this case, the decision boundary points in the gap will get wiped out on all $h \in H$, leaving the exact same $h_\Delta$. Because $H$ is nearly unrestricted in the range $-k \leq x_1 \leq k$, it is capable of shattering infinite points that lie in this range and has infinite VC dimension. On the other hand, because $|H_\Delta| = 1$, we get $\text{VC}(H_\Delta) = 0$. □

### B.9 Proof of Theorem 2

**Definition 1.** *A hypothesis class $H$ is said to be* closed to input scaling *if for all $h(x) \in H$ and for all $a > 0$, we have that $h_a(x) = h(ax) \in H$ as well.*

Note that the families of polynomials and polytopes are clearly closed to input scaling. Furthermore, most classes of neural networks are as well since $h(ax)$ can be realized by scaling the first weight vector by $a$.

**Theorem 2:** *If $H$ is closed under input scaling and $\text{VC}(H) < \infty$, then $\text{VC}(H) \leq \text{VC}(H_\Delta)$.*

*Proof.* Intuitively, the reason that the VC dimension cannot decrease is that for any $H$ with a VC of dimension $k$ that is closed to input scaling, we can find a set $S$ of $k$ points that can be shattered by $H$, where none of the points $s \in S$ strategically move under any of the $h \in H$ used to shatter $S$. As a result, $H_\Delta$ will also shatter $S$, so its VC dimension is at least $k$.

Let $\text{VC}(H) = k$ and $G \subset H$ be a set of $2^k$ classifiers that shatters some set $S$ of $k$ points. Define $c_{DB}(s, g)$ as the minimum strategic cost necessary for point $s$ to obtain a positive classification from classifier $g$ and define $c_{neg}(S, G)$ as the minimum strategic cost for any $s \in S$ to obtain a positive classification from a classifier $g \in G$ that classifies $s$ negatively:

$$c_{neg}(S, G) = \min_{s \in S, \, g \in G, \, g(s)=0} c_{DB}(s, g) \tag{5}$$

Since both $S$ and $G$ are finite sets and negative points face a strictly positive cost to obtain a positive classification, we get that there exists a unique value $c_{neg}(S, G) > 0$. Additionally, because the VC dimension is indifferent to input scaling[5], if we scale both the classifiers and the points, then $G' = \{g(ax) : g(x) \in G\}$ will shatter $S' = \{ax : x \in S\}$ with $c_{neg}(S', G') = a \cdot c_{neg}(S, G)$. Note that because $H$ is closed to input scaling, $G' \subset H$ as well. Furthermore, if we choose $a = \frac{\alpha+1}{c_{neg}(S,G)}$, then $c_{neg}(S', G') = \alpha + 1$. Note that if $c_{neg}(S', G') > \alpha$, no point $s' \in S'$ will move strategically under any $g' \in G'$, meaning that $g'_\Delta(s') = g'(s') \; \forall g' \in G, \; s' \in S'$. As a result, $G'_\Delta$ will also shatter $S'$ so $\text{VC}(G'_\Delta) \geq |S'| = k$. Since $G'_\Delta \subset H_\Delta$, we get that

$$\text{VC}(H_\Delta) \geq \text{VC}(G'_\Delta) \geq k = \text{VC}(H)$$

$\square$

## B.10 Proof of Theorem 3

**Theorem 3:** *Let $H^{m,k}$ be a class of piecewise-linear classifiers with at most m segments and k intersections of linear segments. Then $\text{VC}(H_\Delta^{m,k}) = O\big(d\,m\,log(m) + \nu_p\,k\,log(k)\big)$, where $\nu_p$ is the VC of the class of $\ell_p$ norm balls, and is $\Theta(d)$ for $p = 1, 2, \infty$.*

*Proof.* We will make use of the following known results:

**Lemma 2** (Gómez and Kaufmann (2021)). *Let $H^{Ball}$ be the class of all $\ell_p$ balls in $R^d$. Then $H^{Ball}$ has VC dimension $\text{VC}(H^{Ball}) = \Theta(d)$ for p = 1,2, or $\infty$ [15].*

**Lemma 3** (Vaart and Wellner (2009)). *Let classes $C_1, C_2, ..C_k$ have VC dimensions $V_1, V_2, ...V_k$ and $V = \sum_{i=1}^{k} V_i$. Let $C_U = \bigsqcup_{j=1}^{m} C_j \equiv \{\bigcup_{j=1}^{m} h_j : h_j \in C_j\}$, and let $C_I = \prod_{j=1}^{m} C_j \equiv \{\bigcap_{j=1}^{m} h_j : h_j \in C_j\}$. Then both $C_U$ and $C_I$ have VC dimension upper bounded by $O\big(V \cdot log(k)\big)$ [34].*

We note that by combining Lemmas 2 and 3, the class $H_k^{Ball}$ of the union of up to $k$ $\ell_p$ balls has a VC dimension of $O\big(\nu_p\,k\,log(k)\big)$, which becomes $O\big(d\,k\,log(k)\big)$ in the case of either the $\ell_1, \ell_2$, or $\ell_\infty$ norm cost functions. Additionally, the class $H_m^{Lin}$ of piecewise linear models with up to $m$ segments has a VC dimension of $O\big(d\,m\,log(m)\big)$. [6]

Because linear classifiers map to new shifted linear classifiers, all we have to do to determine the effective classifier of a piecewise-linear $h$ is to combine (take the union of) the effective segments of each linear segment in $h$ and the potential $\ell_p$ ball artifacts around each intersection. Note that, as seen in Fig. 7 (Left), some of the effective segments may be wiped out in this process, and, as seen in Fig. 7 (Right), only intersections that open to the positive region will induce $\ell_p$ ball artifacts in $h_\Delta$. Therefore, the $h_\Delta$ of a piecewise linear $h$ with $m$ segments and $k$ intersections will be the intersection of a piecewise linear classifier $g$ with up to $m$ segments and up to $k$ $\ell_p$ ball artifacts. As a result, we can write $H_\Delta \subseteq G \cap H_k^{Ball}$, where $G \subseteq H_m^{Lin}$. Therefore:

---

[5]Input scaling only changes the distance between points, but does not change their relative positions, so the class $H$ and its input-scaled class $H'$ will have the same VC dimension.

[6]Note that combining the two technically proves the bounds for the classes of the union of *exactly k* balls and *exactly m* segments, respectively, though we can easily expand them to the classes of *up to k* balls and *up to m* segments by recognizing that some of the balls/segments from classes $C_1, C_2, ...C_k$ may be the exact same.

$$\mathrm{VC}(H_\Delta) \leq \mathrm{VC}\big(G \cap H_k^{ball}\big)$$
$$= O\big(\mathrm{VC}(G) + \mathrm{VC}(H_k^{ball})\big)$$
$$\leq O\big(\mathrm{VC}(H_m^{Lin}) + \mathrm{VC}(H_k^{ball})\big)$$
$$= O\big(d\,m\,log(m) + \nu_p\,k\,log(k)\big)$$

Consequently, if $H$ is learnable, then $H_\Delta$ is too.

$\square$

## B.11 Proof of Theorem 4

**Theorem 4:** *Consider either the $\ell_1$, $\ell_2$, or $\ell_\infty$ costs and let $H \subseteq P^k$, where $P^k$ is the set of all $k$-vertex polytopes in $R^d$ and has $\mathrm{VC}(P^k) = O(d^2\,k\,\log k)$[23]. Then $\mathrm{VC}(H_\Delta) = O(d^2\,k\,\log k)$.*

*Proof.* As seen in Appendix B.6, the effective classifier of a k-vertex polytope is the intersection of a larger radius (or smaller for negative inside polytopes) k-vertex polytope and $k$ $\ell_p$ balls centered at each of the vertices. Therefore, $H_\Delta \subset P^k \cap H_k^{ball}$, so by Lemmas 2 and 3 (and the VC dimension of $H_k^{ball}$ derived in Appendix B.10):

$$\mathrm{VC}(H_\Delta) \leq \mathrm{VC}\big(P^k \cap H_k^{ball}\big)$$
$$= O\big(\mathrm{VC}(P^k) + \mathrm{VC}(H_k^{ball})\big)$$
$$= O\big(d^2\,k\,log(k)\big) + O\big(d\,k\,log(k)\big)$$
$$= O\big(d^2\,k\,log(k)\big)$$

$\square$

## B.12 Proof of Proposition 5

**Proposition 5:** *For all d, there exist (non-degenerate) distributions on $\mathbb{R}^d$ that are realizable in the standard setting, but whose maximum strategic accuracy is the majority class rate, $\max_y p(y)$.*

*Proof.* Fix $d$ and choose any arbitrary set of positive points $\mathcal{X}_1$. When $d = 1$, the set of negative points $\mathcal{X}_0$ can be constructed by placing a negative point $\delta$ to the left and right of each positive point. When $d > 1$, this generalizes to selecting $d + 1$ points around each positive point $x \in \mathcal{X}_1$ that form an $\mathbb{R}^d$ simplex $T_\delta^d(x)$[7]. In this case, the classifier $h(x) = \mathbb{1}\{x \in \mathcal{X}_1\}$ achieves perfect standard accuracy, and the classifier that classifies all points as negative will achieve a strategic accuracy of $\frac{d+1}{d+2}$. Note that for each point $q$ in space to which $x \in \mathcal{X}_1$ can strategically move, there exists at least one $x' \in T_\delta^d(x)$ that can as well. Therefore, for each positive point $x \in \mathcal{X}_1$ that achieves a correct positive classification by strategically moving to $q$ ($q$ can be the same as $x$ if it doesn't move), there is a unique negative point $x' \in T_\delta^d(x)$ that achieves an incorrect negative classification by strategically moving to $q$, so the strategic accuracy does not benefit from correctly classifying any positive points. As a result, the maximum strategic accuracy is achieved by correctly classifying each negative point and incorrectly classifying each positive point, which gives the majority class rate.

$\square$

## B.13 Proof of Proposition 6

**Proposition 6:** *There exists a distribution that is realizable in the standard setting, but whose accuracy under any $h_\Delta$ is at most $0.5 + \frac{1}{\lfloor 2\alpha+1 \rfloor}$.*

---

[7]We can assume without loss of generality that $\bigcap_{x_1, x_2 \in \mathcal{X}_1} T_\delta^d(x) = \varnothing \ \forall x_1, x_2 \in \mathcal{X}_1$ since there are infinite options for each $T_\delta^d$, and we can just pick ones that do not overlap.

*Proof.* Consider the set of lattice points in $\mathbb{R}^1$ with alternating labels. Although $f(x) = sin(\pi x)$ achieves perfect standard accuracy on this distribution, we will show that the maximum strategic accuracy for any model class is $0.5 + \frac{1}{\lfloor 2\alpha+1\rfloor}$, which approaches 0.5 as $\alpha \to \infty$. Note that in $\hat{R}^1$, the minimum length of a positive interval is $2\alpha$. Additionally, for each positive interval $I$ of length $k$, $I$ will contain either $\lfloor k \rfloor$ or $\lfloor k \rfloor + 1$ points and include at most one more positive point than negative points. Therefore, each interval $I$ will, at best, be correct on $\frac{\lfloor k \rfloor + 1}{2}$ out of $\lfloor k \rfloor$ points, which is optimized for $k = 2\alpha$ (i.e., the minimum possible value of $k$). Furthermore, because the negative intervals may also only have at most one more negative point than positive points, the negative intervals are optimized when they include only a single negative point. As such, an optimal strategic classifier $h$ consists of repeated positive intervals that include $\lfloor 2\alpha \rfloor$ points, followed by a negative interval that includes one negative point. For each positive and negative interval group, $h$ correctly classifies at most $\frac{\lfloor 2\alpha \rfloor + 1}{2} + 1$ out of $\lfloor 2\alpha + 1 \rfloor$ points correctly for an overall maximum accuracy of $\frac{\frac{\lfloor 2\alpha+1 \rfloor}{2}+1}{\lfloor 2\alpha+1 \rfloor} = 0.5 + \frac{1}{\lfloor 2\alpha+1 \rfloor}$. $\qquad\square$

## C   Additional results

### C.1   Other Cost functions

Our low-level analysis (Sec. 4) begins with a focus on the $\ell_2$-norm cost, both because it is the most popular choice in the literature, and because we found it to offer the best intuition. Nonetheless, most of our results hold more broadly, and in particular, all theorems and propositions in Sec. 5 generalize to any $\ell_p$ ($p \geq 1$) norm:

- Prop. 4 applies generally, since negative-inside polytopes are still mapped to smaller negative-inside polytopes under general $\ell_p$ cost functions.

- Thm. 1, Cor. 1, and the example in Fig. 4 of increasing VC dimension remain under minor tweaks to the constructions that take into account the differences in shapes of $\ell_p$ balls for different values of $p \geq 1$.

- Thm. 2 holds for any cost function under which the maximum Euclidean distance any user can strategically move is bounded by a constant. This holds not only for $\ell_p$ norms, but also for most reasonable cost functions.

- Thm. 3 is already stated for general $\ell_p$ norms.

- Thm. 4 is stated for $\ell_1$, $\ell_2$, and $\ell_\infty$. However, a more general result can be stated with dependence on the VC dimension of the cost function balls (i.e., the set $C = \{c(x, x') \leq \alpha \; \forall x \in \mathbb{R}^d\}$). The bound would then be $O(d^2 \, k \, log(k) + \text{VC}(C))$, and would support any plug-in results for $\text{VC}(C)$.

- Prop. 5 and Prop. 6 also remain under minor tweaks to the constructions that take into account the differences in shapes of $\ell_p$ balls for different values of $p \geq 1$.

- Obs. 3, 4, 5, and Cor. 2 hold for $\ell_p$ norms in general as well, since they are not shape dependent, but rather results of movement within a ball.

Thus, our claims regarding the fact that non-learnable classes can become learnable, the loss of universality, and the limits to approximation all hold more generally.

In terms of asymmetric norms such as the Mahalanobis norm, while these norms would complicate the curvature analysis (and subsequent impossibility results that build off them), all results in Sec. 5 would still hold. This is because while the shape of balls changes (like with other $\ell_p$ balls), fundamental properties like lines mapping to other lines do not. As such, the results in Sec. 5 would still hold for the exact same reasons as delineated above for $\ell_p$ balls.

As for feature-dependent (also known as "instance-wise") costs, these are qualitatively different from the conventional global costs, in a way that can make the claims become irrelevant or even degenerate. For example, Sundaram et al. [33] show that on the one hand, even for linear $H$, instance-wise cost function can cause $\text{VC}(H_\Delta) = \infty$, and on the other hand, for separable costs (which are instance-wise) it holds that $\text{VC}(H_\Delta) \leq 2$ for any $H$.

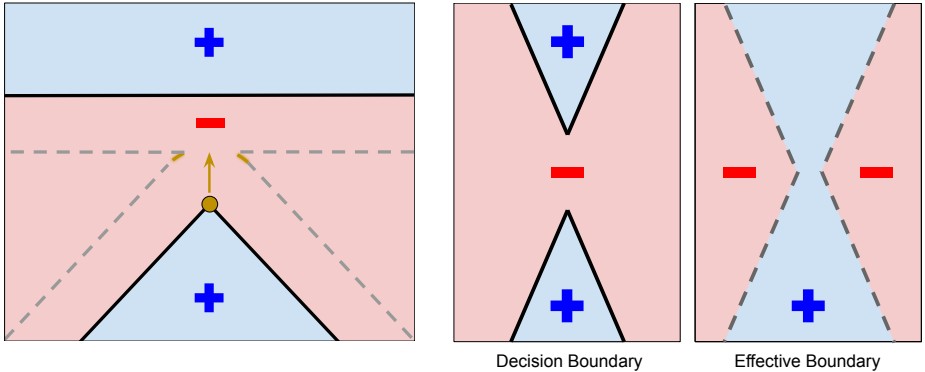

Figure 9: **(Left)**: Partial expansion. **(Right)**: Regions merging/splitting

## C.2    Partial Expansion

As seen in Fig. 9 (Left), partial expansion may occur if part of the expansion artifact is discarded through *indirect wipeout*. In the figure, part of the artifact (gold) is within $\alpha$ of the upper positive region and therefore does not become part of the decision boundary.

## C.3    Merging/Splitting Regions

As seen in Fig. 9 (Right), two positive regions can merge together when they are separated by a narrow pass in $h$. If the narrow pass is part of a larger negative region, the region will be split into two, which can potentially increase the complexity of the decision boundary (for example, in Fig. 9 (Left)).

## C.4    Types of Impossible Effective Classifiers

As stated in Sec. 4.4, we identify four "types" of classifiers $g$ that cannot exist as effective classifiers. Formally, these classifiers are ones which include either:

1. A small continuous region of positively classified points $C$ where $\exists q \in R^d$ such that $x \in B_\alpha(q) \, \forall x \in C$.

2. A narrow continuous region $C$ where all points are classified as positive and are $\alpha$-close to the decision boundary.

3. A smooth point on the decision boundary which has signed curvature $\kappa > 1/\alpha$ (or $\kappa^* > 1/\alpha$ when $d > 2$).

4. A locally convex intersection of piecewise linear or hyperlinear segments.

Note that the first case is just a special case of the second case. Nonetheless, we have separated them to emphasize that the impossible positive regions can either be fully or partially contained.

### Cases 1 and 2: Small/Narrow Positive Regions

Let $x \in C$ be some point in a small positive region in $g$. For $x$ to be classified as positive, there must be a point $x' \in B_\alpha(x)$ which is classified positively by $h$ and not reachable by any $q \in \mathcal{X}_0(h)$ (see requirements 1 and 2 in Appendix B.4). However, any point $x' \in B_\alpha(x) \cap C$ is reachable by $\mathcal{X}_0(h)$ and any point $x' \in B_\alpha(x) \cap C^c$ is also reachable by $\mathcal{X}_0(h)$ since it is closer to some $q \in \mathcal{X}_0(h)$ than to $x$. Therefore, there are no points $x' \in B_\alpha(x)$ which are not reachable by any $q \in \mathcal{X}_0(h)$, so $g$ cannot be an effective classifier.

### Case 3: Large positive curvature

We will show that any $g$ with directional curvature greater than $1/\alpha$ anywhere on the decision boundary will fail Prop. 3. In other words, if the directional curvature at point $x$ is greater than $1/\alpha$, then the point $x + \alpha \hat{n}_x$ will be reachable by some point that is negatively classified by $g$. In order for the point $x + \alpha \hat{n}_x$ to be reachable by $x$ but not any point negatively classified by $g$, the

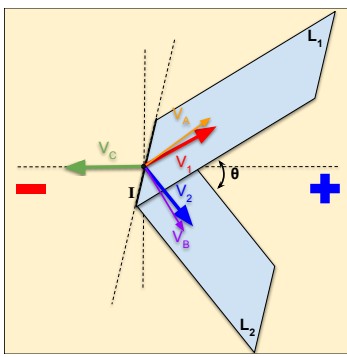

Figure 10: Diagram for proof of case 4.

ball $B_\alpha(x + \alpha\hat{n}_x)$ — which is the set of points that can strategically move to $x + \alpha\hat{n}_x$ — must not intersect $g$ and must also be strictly tangent to it at $x$. However, if the curvature at point $x$ is greater than $1/\alpha$, then the maximum radius $r_{insc}$ of a ball that is strictly tangent to $x$ is less than $\alpha$, so $g$ does not meet Prop. 3.

In $\mathbb{R}^2$, Lemma 1 proves that the radius $r_{insc}$ of the maximum inscribed circle at $x$ is no bigger than the radius $r_{osc} = \frac{1}{\kappa}$ of the osculating circle at $x$. We therefore have that $r_{insc} \leq r_{osc} = \frac{1}{\kappa} < \alpha$. In higher dimensions (see Appendix A.1), there is no longer a single normal curvature. Instead, we define the set $K$ of curvatures in each plane $p \in P$ defined by the normal vector and some tangent vector at $x$. Note that $B_\alpha(x + \alpha\hat{n}_x)$ is only strictly tangent to the decision boundary at $x$ if it's intersection with each $p \in P$ (a circle) is strictly tangent at $x$ to the curve obtained by intersecting the decision boundary with the plane $p$. Since each of these cases are in $\mathbb{R}^2$, each one will only hold if the curvature at $x$ in $p$ is no more than $1/\alpha$. Therefore, if $\kappa^* = \sup(K) > 1/\alpha$ for any $x$ on the decision boundary, then the maximum radius of a ball that is strictly tangent to $x$ is less than $\alpha$, so $g$ does not meet Prop. 3.

**Case 4: Piecewise Linear Segments**

**Lemma 4.** *The set of points that are reachable by point $x$ but not point $x + \delta\vec{v}$ for unit vector $\vec{v}$ and $\delta \to 0$ all satisfy $\vec{v} \cdot x \leq 0$.*

*Proof (Lemma 4).* Define an orthonormal basis $v_1, v_2...v_d$ with $v_1 = v$. For any point $q$ such that $\vec{v} \cdot q > 0$, the vector connecting $x$ and $q$ can be written as $a_1 v_1 + a_2 v_2... + a_d v_d$ while the vector connecting $x + \delta\vec{v}$ and $q$ can be written as $(a_1 - \delta)v_1 + a_2 v_2... + a_d v_d$. Since $0 < \delta < a_1$ (note that $\vec{v} \cdot q > 0$ so $a_1 > 0$), $q$ is closer to $x + \delta\vec{v}$ than $x$. Therefore, any point that is reachable by $x$ and not by $x + \delta\vec{v}$ must satisfy $\vec{v} \cdot x \leq 0$. $\square$

*Proof (Case 4).* As depicted in Fig. 10, let $L_1$ and $L_2$ be two $R^d$ hyperplane segments that intersect on the decision boundary of $g$ at the $R^{d-1}$ hyperplane segment $I$, and let $x$ be some point on I. Let $0 < \theta < \pi$ be the angle between $L_1$ and $L_2$ on the positive side. Let $\vec{v}_1$ be the unit vector in $L_1$ that runs through $x$ and is orthogonal to $I$; $\vec{v}_2$ be the unit vector in $L_2$ that runs through $x$ and is orthogonal to $I$; and $\vec{v}_C = -\frac{\vec{v}_1 + \vec{v}_2}{||\vec{v}_1 + \vec{v}_2||}$. Note that for $\delta \to 0$ the points $x_1 = x + \delta\vec{v}_1$ and $x_2 = x + \delta\vec{v}_2$ are on the decision boundary while the point $x_{12} = x + \delta\vec{v}_C$ is classified as positive and $x_C = x - \delta\vec{v}_C$ is classified as negative. Additionally, if we define $\vec{v}_A$ as the the result of $\vec{v}_1$ being rotated $0 < \epsilon < \frac{\pi - \theta}{2}$ around $I$ away from the positive region and $\vec{v}_B$ as the result of $\vec{v}_2$ being rotated $\epsilon$ around $I$ away from the positive region, then both $x_A = x + \delta\vec{v}_A$ and $x_B = x + \delta\vec{v}_B$ are classified as negative.

Assume that there exists a classifier $h$ associated with the effective classifier $g$. Therefore, there exists a point $q$ that is reachable by $x$ but not by $x_A$, $x_B$, or $x_C$ (which are classified as negative by $g$). By Lemma 4, any point $q$ that is reachable under strategic movement by point $x$ but not $x_A$, $x_B$, or $x_C$ must at least satisfy $q \cdot \vec{v}_A \leq 0$, $q \cdot \vec{v}_B \leq 0$, and $q \cdot \vec{v}_C \leq 0$. However, because $\vec{v}_A + \vec{v}_B$ points in the same direction as $\vec{v}_1 + \vec{v}_2$ — and therefore the opposite direction of $\vec{v}_C$ — we must have that $q \cdot \vec{v}_C = 0$ and $q \cdot (\vec{v}_A + \vec{v}_B) = 0$, which implies that both $q \cdot \vec{v}_A = 0$ and $q \cdot \vec{v}_B = 0$. However, this implies that the vector $\vec{q}$ is orthogonal to both $\vec{v}_A$ and $\vec{v}_B$, which can only happen if $q$ lies on $I$ or if the angle between $\vec{v}_A$ and $\vec{v}_B$ is either 0 or $\pi$. In the former case, $q$ is then reachable

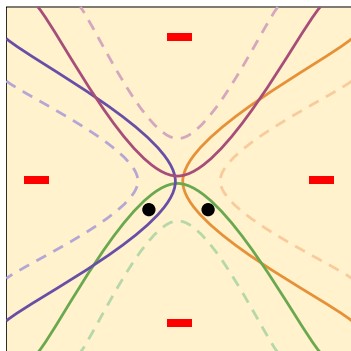

Figure 11: Example of VC decreasing from 2 to 1. Each color represents a different classifier (drawn together to illustrate shattering capacity), and the dashed curves represent the corresponding effective classifiers. Though the black points can be shattered by $H$, none of the $h_\triangle$ overlap so $\mathrm{VC}(H_\triangle) = 1$.

by the negatively classified point that is just over the decision boundary from it. The latter case is also not an option because the angle between $\vec{v}_A$ and $\vec{v}_B$ is $\theta_{AB} = \theta + 2\epsilon$, and we have ensured that $0 < \theta + 2\epsilon < \theta + 2\frac{\pi-\theta}{2} = \pi$. In either case, it is impossible for $g$ to classify $x$ as positive while classifying $x_A$, $x_B$, and $x_C$ as negative, so there is no $h$ such that $g = h_\triangle$. $\qquad\square$

## C.5  Non-bijectiveness of Classifier Mapping

In Sec. 4.4, we show that the mapping $h \mapsto h_\triangle$ is not surjective by showing that there are many potential $h_\triangle$ with no associated $h$. Here, we will show that $h \mapsto h_\triangle$ is also not injective since many classifiers map to the same effective classifier. Furthermore, we show that there are actually infinite $h$ that map to nearly all possible $h_\triangle$. Though there are many more interesting examples, the simplest case of non-injectiveness can be seen by considering an $h$ that includes a continuous positive region $C$ with an infinite number of points. Define the classifier $h^x$ as the classifier that is identical to $h$ except that $h^x$ classifies $x \in C$ as negative instead of positive. Because the point $x$ will achieve a positive classification through strategic movement, $h^x_\triangle$ is unaffected by $x$ and is the same as $h_\triangle$. Moreover, $h^x_\triangle = h_\triangle$ is true of any $x \in C$, so as long as any $h$ that maps to $h_\triangle$ contains a positive region with infinite points, then $h_\triangle$ will have infinite $h$ that all map to it. Because positive points in $h$ map to positive radius-$\alpha$ balls in $h_\triangle$, any $h_\triangle$ with a positive region $C$ such that $\exists x : \ B_\alpha(x) \subset C$ will have infinite $h$ that all map to it. Note that since all positive regions in $h_\triangle$ have a minimum size requirement $\exists x : \ B_\alpha(x) \subseteq C$, the aforementioned requirement covers all but the $h_\triangle$ with absolute minimum-size positive regions.

## C.6  Classes with $H = H_\triangle$

Though $H \neq H_\triangle$ for most classes $H$, there are a few rather simple $H$ other than the class of linear functions where $H = H_\triangle$. For example, under any $\ell_p$ norm cost, an $\ell_p$ ball with a positive inside will grow by a radius of $\alpha$, while an $\ell_p$ ball with a negative inside will shrink by a radius of $\alpha$ (at least until the radius hits 0). Therefore, many model classes of the subsets of the $\ell_p$ balls — including the class of all negative-inside balls and the class of balls with a radius that is a multiple of $\alpha$ — all have $H = H_\triangle$ for any fixed $\alpha$.

Additionally, since each line segment and $\ell_p$ arc maps to another line segment or $\ell_p$ arc, many piecewise-linear or piecewise $\ell_p$ arc classes will also have $H = H_\triangle$. For example, any concave linear intersection will be mapped to a shifted concave linear intersection, so the class of concave linear intersections satisfies $H = H_\triangle$.

## C.7  Classes with $\mathrm{VC}(H) > \mathrm{VC}(H_\triangle) > 0$

There are a couple of mechanisms that can lead to a decrease in VC dimension. In the first mechanism, multiple different $h$ have the same $h_\triangle$ leading to a smaller and less expressive $H_\triangle$. Two such cases are shown in Appendix B.8 to prove that the VC may decrease even from $\infty$ to 0. In a similar fashion,

we can build an $H$ where only some (instead of all) of the $h \in H$ have the same $h_\Delta$, which would decrease the VC dimension to a non-zero effective VC dimension.

In the second mechanism, the VC can change due to effective classifiers gaining or losing overlap from strategic movement. As the dual to Fig. 4 (Left) where the VC *increases* from 1 to 2 due to this mechanism, Fig. 11 presents an example where the VC *decreases* from 2 to 1. Here, $H$ includes four overlapping classifiers with $\text{VC}(H) = 2$. Yet, under strategic behavior, the effective classifiers cease to overlap, decreasing the shattering capacity to 1.

### C.8   Example of $H_\Delta$ with $\text{VC}(H_\Delta) = \Theta(d \cdot (\text{VC}(H))$

**Claim:**   *There exists a class $H$ with $\text{VC}(H_\Delta) = \Theta(d \cdot (\text{VC}(H))$.*

*Proof.* For simplification of notation, we build on the case shown in Fig. 4 (Left) using classifiers $h$ that classify some ball as positive and all other points as negative. Note that a similar proof can be built based on a class $H$ of point classifiers [7], but we have decided to include the following construction since classifiers that classify only a single point as positive are quite unrealistic. Let set $S = \mathbb{Z}_2^d - \{0^d\} + \{-\delta^d\}$ be the set of $\mathbb{Z}_2^d$ lattice points with the point $\{0^d\}$ replaced by the point $\{-\delta^d\}$. Consider the case where $\alpha = 0.75$ and $H = \{B_{0.25}(x) : x \in S\}$ is the class of radius-0.25 balls around the points in $S$. Because the radius of positive balls expand by $\alpha$ under strategic movement, we get that $H_\Delta = \{B_1(x) : x \in S\}$. Note that because none of the balls in $H$ overlap, $\text{VC}(H) = 1$. On the other hand, the balls in $H_\Delta$ do overlap, allowing $H_\Delta$ to shatter the set of $d$ one-hot points $E = \{e_i : i \in [1...d]\}$, where $e_i$ denotes the point with a 1 in the ith coordinate and 0's elsewhere. Note that for each $x \in S - \{-\delta^d\}$, the ball $B_1(x) \in H_\Delta$ classifies $E$ according to the sign pattern $x$. Additionally, the ball $B_1(-\delta^d)$ does not classify any point in $E$ as positive, so $H_\Delta$ covers all $2^d$ sign patterns of $E$. As such, $\text{VC}(H_\Delta) \geq d$. However, since $H_\Delta \subset B = \{B_1(x) : x \in R^d\}$, we get that $\text{VC}(H_\Delta) \leq \text{VC}(B) = d + 1$ [15] giving $\text{VC}(H_\Delta) = \Theta(d \cdot (VC(H))$. $\square$

### C.9   Example of Strategic Accuracy Exceeding Standard Accuracy

**Claim:**   *In the unrealizable setting, the maximum strategic accuracy can exceed the maximum standard accuracy.*

*Proof.* In the unrealizable setting, the maximum strategic accuracy can exceed the maximum standard accuracy when strategic movement helps correct the incorrect classification of positive points. For example, consider a basic setup where $\alpha = 1$, $X = \{(-1,0),(2,0),(-3,0)\}$, $Y = \{1,1,-1\}$, and $H = \{h_1(x) = sign(x_1 + x_2), h_2(x) = sign(x_1 - x_2 - 1)\}$. In the standard case, both $h_1$ and $h_2$ correctly classify the second and third points, but not the first point, so the maximum accuracy is $\frac{2}{3}$. However, in the strategic setting, the first point is able to obtain a positive classification under $h_1$ by moving to $(-\frac{1}{3}, \frac{2}{3})$ (and the third point still cannot get a positive prediction), so the maximum strategic accuracy is 1. $\square$

### C.10   Practical Takeaways

Given the challenge of optimization in strategic batch learning (even in simpler settings), our paper aims to first establish an understanding of the challenges inherent to non-linear strategic learning (in particular in relation to the linear case), with the hope that our results and conclusions can help guide the future design of learning algorithms, as well as set expectations for what is achievable and what is not. The difficulty of designing a principled algorithm for the general non-linear case is underscored by the fact that existing works rely on strong assumptions to enable optimization. For example, the original paper of Hardt et al. [16] provides an algorithm, but requires assumptions on the cost function that reduce the problem to a one-dimensional learning task, and their algorithm is essentially a line search. Levanon and Rosenfeld [25] differentiate through $\Delta$, but this relies strongly on this operation being a linear projection, which applies only to linear classifiers. Neither of the above naturally extend to the non-linear case.

Towards the goal of practical learning in the general strategic setting, we provide here some examples of how our results can be used as practical takeaways for effective learning:

1. **Learning via inversion:** One implication of our function-level analysis is that for any classifier $h$ with strategic accuracy $\mathrm{acc}_{strat}(h)$, the classifier $h'$ such that $h \mapsto h_\Delta = h'$ has regular (i.e., non-strategic) accuracy $\mathrm{acc}(h') = \mathrm{acc}_{strat}(h)$. Thus, one approach to optimizing strategic accuracy is via reduction: (i) train $h' \in H'$ to maximize *non-strategic* accuracy using any conventional approach for some choice of $H'$, and then (ii) apply the inverse function mapping (which may not be 1-to-1) to obtain a non-strategic classifier $h$, whose effective decision boundary is that of $h'$. This approach requires the ability to solve the 'inverse' problem (which may not be 1-to-1) to find the effective classifier. We discuss cases where this would and wouldn't work in Sec. 4.4.

2. **Regularizing curvature:** A sufficient condition for a pre-image strategic $h'$ to exist for a given non-strategic $h$ is that the function mapping is 1-to-1. Rather than restrict learning to only invertible classes $H'$ a priori, an alternative approach is to work with general $H'$ but regularize against those $h' \in H'$ that do not have an inverse. Prop. 1 implies that one way to promote this is by encouraging $h'$ that are smooth, since low-curvature classifiers are less prone to *direct wipeout*, and therefore more likely to permit inversion.

3. **Finding the interpolation threshold:** The interpolation threshold — the point in which the number of model parameters (or model complexity more generally) attains minimal training error — is key for learning in practice: it marks the extreme point of overfitting (in the classic underparametrized regime) and the beginning of benign overfitting (in the modern overparametrized regime, e.g., of deep neural nets). In non-strategic learning, this point is easily identified as that where the training error is zero. In contrast, our results show that in strategic learning, the minimal training error can be strictly positive. This makes it unclear when (and if) the threshold has been reached. Fortunately, our procedure for our approximation experiment in Sec. 6.2 can be used generally to compute the minimal attainable strategic training error, independently of the chosen model class. This provides a tool that can be used in practice to measure interpolation.

In terms of model class selection, our results in Sec. 5 offer insight into potential upsides and downsides to certain classes. One implication of our results is that strategic behavior can generally either increase or decrease the complexity of the chosen model class. However, Thm. 2 ensures that the VC dimension can only increase for any class that is closed to input scaling, which includes many common classes used in practice. Thus, a learner who chooses such classes is guaranteed that the potential expressivity of the learned classifier will not be deteriorated by strategic behavior. Additionally, Prop. 4, Thm. 4, and Thm. 3 all give upper bounds on the SVC of particular classes (e.g., ReLU neural networks). The implication is that a learner choosing to work with such $H$ is guaranteed that if $H$ is learnable, then the induced $H_\Delta$ is also learnable. Finally, Obs. 5 and Thm. 1 both point to the fact that several non-strategic classifiers $h$ can be mapped to the exact same effective classifier $h_\Delta$. If a class $H$ includes many such cases, then it has significant redundancy, which can have negative implications on the process of choosing a good $h \in H$ via learning (e.g., in terms of generalization, optimization, or the effectiveness of proxy losses). Ideally, the learner should be cautious of classes in which this is prevalent, for example, classes that permit high curvature or small negative areas.

# D    Experimental details

## D.1    Experiment #1: Expressivity

In Sec. 6.1, we experimentally demonstrate how the strategic setting affects classifier expressivity. In this experiment, we first randomly sample a degree-$k$ polynomial $h$, then determine the set of points $S$ that make up the decision boundary of $h_\Delta$ (i.e., are directly adjacent to it), and finally find the best approximate polynomial fit of $S$ [8]. Because user features are often naturally bounded, we assume that $x \in \mathcal{X} = \mathbb{R}^2$ and that all $x$ are bounded by $|x|_\infty \leq 100$ (see Appendix A.2 for a discussion on how our theoretical results hold under bounded input domains). We report average values and

---

[8]Because we are concerned specifically with the polynomial fit of $h_\Delta$, we label the points in a fine grid G both regularly and then strategically, and use the strategic labels to get a polynomial fit. In this way, we do not directly compute the effective classifier (which would be difficult), but approximate it well enough to get the polynomial fit as well as visualize it.

standard error confidence intervals of 100 instances of the setup for each $\alpha \in \{2, 4, 8, 16, 24, 32, 40\}$ and $k \in [1, 10]$. The instances were divided among 100 CPUs to speed up computation.

To randomly sample polynomials of degree $k$, we leverage the fact that the number of points needed to uniquely determine a degree-$k$ polynomial over $\mathbb{R}^2$ is $\binom{k+2}{2}$. Therefore, for each instance, we randomly choose $\binom{k+2}{2}$ points and labels, and use an SVM polynomial fitter to find the best fitting degree-$k$ polynomial fit to these points. To ensure that the generated polynomial is indeed a full-degree polynomial, we construct a grid $G$ of points over $[-100, 100]^2$, label each point $g \in G$ using $h$ to get labels $Y$, and verify that the SVM best-fitting degree-$k$ polynomial achieves near-perfect accuracy on $(G, Y)$, while the best $(k-1)$-degree polynomial achieves low accuracy on $(G, Y)$.

To find the set of points $S$ that make up the decision boundary of $h_\triangle$, we first strategically label each $g \in G$ by checking if $g$ can reach any point $g'$ that is classified as positive by $h$. To determine the points that make up the decision boundary of $h_\triangle$, we select only the points $g \in G$ where $g$ and at least one of its direct neighbors are labeled differently by $h_\triangle$.

Finally, we again use an SVM polynomial fit to determine the best approximate polynomial degree of $h_\triangle$. Because the $h_\triangle$ of a polynomial is not necessarily a polynomial itself, we set $k'$ to be the lowest degree of the polynomial whose fit on $S$ passes a set tolerance threshold. Empirical tests showed that a tolerance of 0.9 was high enough to ensure a good fit on all of $G$, but not so high that the SVM fit algorithm needlessly increased the degree just to get a complete overfit to $S$.

## D.2 Experiment #2: Approximation

In Sec. 6.2, we experimentally demonstrate the effects of non-universality on the maximum strategic accuracy. This experiment consists of (i) sampling synthetic data and (ii) calculating the maximum linear accuracy and strategic accuracy on each dataset instance. We report average values and standard error confidence intervals of 20 instances of the setup for each $\alpha \in \{0.5, 1, 2\}$ and $\mu \in [0, 5]$. The instances were divided among 100 CPUs to speed up computation.

For each instance, we generate synthetic $\mathbb{R}^2$ data by sampling points from class-conditional Gaussians $x \sim N(y\mu, 1)$, where $\mu$ is the separation between the centers of each class. Because we were unable to find a polynomial-time algorithm to calculate the maximum accuracy of $H_\triangle^*$, exactly 25 points were drawn from each class (denote the full dataset $\mathcal{X}$). Though this setup may not reflect all possible data distributions, it does represent how the strategic environment behaves under increasing data separability.

To calculate the maximum linear accuracy in each instance, we first note that there exists an optimal linear classifier that runs through (or infinitesimally close to for negative points) exactly two dataset points. This is because for any optimal classifier $h$ that runs through exactly one dataset point $x \in \mathcal{X}$, $h$ may be rotated until it runs through another $x' \in \mathcal{X}$ without changing any of the labels. For any optimal classifier $h$ that runs through no $x \in \mathcal{X}$, $h$ can be shifted until it runs through one (or two collinear) $x \in \mathcal{X}$ and then rotated appropriately. Because all data points are random, we assume that no three points are collinear. Therefore, the maximum linear accuracy on $\mathcal{X}$ can be calculated by taking the maximum accuracy over the $\binom{n}{2}$ classifiers that run through each pair of $x, x' \in \mathcal{X}$ [9].

When calculating the maximum strategic accuracy, we note that the minimum size of a positive region in $h_\triangle$ is the ball $B_\alpha$, so any strategic classifier can be represented by the intersection of multiple $B_\alpha$. We therefore calculate the set $S$ of all possible subsets $s \subseteq \mathcal{X}$ that any $B_\alpha$ classifies as positive, and find the intersection $\bigcap_{s \in S' \subseteq S} s$ with the highest accuracy. To find all $s \in S$, we iterate through the circles centered at each point on a fine-grained grid over the range of the dataset (padded by $\alpha$ on all sides) and check which points $x \in \mathcal{X}$ each circle includes. To find the best intersection of these circles, we iterate through each of the subsets of $Z \subseteq \mathcal{X}_0$ [10] and then $s \in S$ to find the maximum number of positive points that can be classified as positive if we allow ourselves to mistakenly classify the negative points in $Z$.

---

[9]Because points on the decision boundary are labeled as positive, we manually define that negative points used to define the linear classifier are accurately labeled to reflect the fact that the optimal classifier does not run through the point, but infinitely close to it.

[10]Subsets are traversed in size order with early stopping once the size of the subset is large enough that the accuracy will be worse than the current best accuracy, even if all positive points are correctly classified.

