# OpenReview forum: "Strategic Classification with Non-Linear Classifiers"
_NeurIPS.cc/2025/Conference — NeurIPS 2025 poster_

### Official Review · Reviewer_qnCe · 2025-06-30

**Clarity:** 3
**Significance:** 2
**Originality:** 3
**Rating:** 4
**Confidence:** 4

**Summary:**

This paper studies strategic classification in the offline geometric setting where the true underlying classifier is nonlinear. The authors take a "bottom up" approach that first considers changes to individual points on the boundary, then aggregate them to the effective classifier under manipulation and subsequently the effective model class. For individual points $x$, they discussed four cases of the mapping from $x$ to the set of points that manipulate to $x$ at maximum cost: one-to-one, wipeout, expansion and collision. For the effective classifiers, they identified two mechanisms that can limit their expressivity --- curvature and containment --- and derive some conditions about effective classifiers that cannot exist. The authors then move on to the model class level and compares the strategic VC dimension and the original one, and show that universal approximators may fail to capture the samples after manipulation.

**Questions:**

Could you comment on how the results in section 6 help the learner to choose model classes in practice?

**Ethical Concerns:**

["NO or VERY MINOR ethics concerns only"]

**Final Justification:**

The rebuttal has addressed my concerns about limited model class-level results by providing a new characterization for piecewise linear functions, and a detailed discussion of the practical implications for selecting hypothesis classes under strategic manipulations. Overall, I appreciate the contributions and decided to raise my score from 3 to 4.

**Limitations:**

yes

**Quality:**

2

**Strengths And Weaknesses:**

Strengths:

The authors systematically study strategic classification with nonlinear classifiers in the continuous geometric setting where agents can move in a bounded radius ball. In contrast, existing work either studies non-linear classifiers in a discrete setting via manipulation graphs, or exclusively focus on linear classifiers in the geometric spaces. The paper is clearly written, nicely builds up intuition via a bottom-up approach, and comprehensively study the structure and complexity of this problem.

Weaknesses:

1. Limited technical depth. While the paper presents many results, most of them feel like simple observations that have been written down formally, rather than deep technical insights. Since the paper takes a bottom-up approach, the results about individual points and classifiers are mostly just listing observations and examples --- this does help build intuition but doesn't really tell us anything new. The most interesting part is the model class-level results in Section 6 (which I consider to be the meat of this paper) where they quantitatively study the complexity and expressivity of the effective model class versus the original ones. But this analysis is incomplete since they only make a conjecture about the VC gap in the general case, but did not provide a satisfactory answer.

2. The authors study both VC dimension/sample complexity and universality of model classes, but these results feel mostly pessimistic and do not provide clear guidance on how to select model classes or design learning algorithms in practice. The experimental results in section 7.2 are also somewhat unsurprising: showing improved accuracy with bigger separation between positive and negative examples and smaller manipulation radius, with H^{lin} being a lower bound. Overall, the practical implications and takeaways from this paper is a bit unclear.

---

> ### Author Rebuttal · Authors · 2025-07-31
>
> Thank you for your thorough review, we appreciate your constructive feedback. Based on your review, we have added (i) a list of practical take-aways based on our analysis, to be added as a new section, and (ii) a new result that extends Thm. 3 to show our conjecture holds for classes of piece-wise linear functions (e.g., ReLU networks). We hope these and our answers to your questions help address your concerns.
>
>
> **”While the paper presents many results, most of them feel like simple observations that have been written down formally, rather than deep technical insights.”**
> The primary goal of Secs. 4 and 5 is to build intuition towards the class-level analysis in Sec. 6 (which, as you note, includes the core of our results). Nonetheless, we believe that our results in the these earlier sections, albeit simple, still hold value in their own right: the non-universality of strategic learning, the fact that some points have no influence on the learned classifier, the ability of strategic behavior to both promote and suppress complexity, and the mechanisms underlying these phenomena and structural claims are all fundamental to understanding the non-linear strategic setting.
> Our goal in this work was to provide a compelling story of how non-linear strategic learning can be qualitatively different from the linear case. We are aware that in this sense our paper deviates from the structure of a conventional theory paper (where emphasis is typically on the depth of technical results). Nonetheless, we believe our results can still be of interest to the community and of value towards future research. We do not see why their simplicity takes away from their significance.
> At the same time, and after reading your review and others, we agree that showing how our results transfer to concrete implications for learning can help better establish the value of our results. **We will therefore add to the paper a new section on practical take-aways** in the next revision using the extra page. We detail such implications in regards to Sec. 6 in our response to your question below. Please also see our response to Rev. YJeu for detailed practical take-aways from other parts of the paper.
>
> **”The most interesting part is the model class-level results in Section 6… But this analysis is incomplete since they only make a conjecture about the VC gap in the general case.”**
> The gap remains a conjecture not for lack of trying (we have thought long and hard about this), but because a generic result has proven to be quite elusive to prove (or disprove). A key reason is that in a strategic setting points will move differently when facing different $h$ from the same $H$. Thus, reasoning about the shattering of an arbitrary set must account for how points can move under *all possible* $h \in H$, though not simultaneously – which proved to be highly challenging in the general, unstructured case. This is why we have included two results that exploit structure: the explicit upper bound on polytopes (Thm. 3), and a lower bound for any $H$ that is closed to scaling (Thm. 2), which includes many common classes used in practice.
> While closure to scaling did not provide us sufficient structure for an upper bound, in response to your concern, **we add here a new result that extends Thm. 3 to show our conjecture holds for any class of piecewise-linear classifiers with a bounded number of segments** – a large and highly expressive set of classes, which includes neural networks with ReLU activations (of fixed size) as a special case, and is often used in studies on neural network approximation. The proof technique is similar to that of Thm. 3 where each linear segment maps to a new linear segment and each intersection can map to a partial to the cost function shape. Each classifier $h_\Delta \in H_\Delta$ is therefore the intersection of a new piecewise linear classifier $g \in H$ (with the same number of segments as the original $h$, but each segment is shifted away by $\alpha$) and a finite (and bounded) number of $L_p$ balls. Note that H has finite VC because each $h \in H$ is the union of $m$ linear classifiers (see Thm. 1.1  in Vaart and Wellner [2009] VC bound of unions) and $H_k^{Ball} $, the class of all classifiers that are the union of the union up to $k$ $L_p$ balls, does too as shown in the proof for Thm. 3. By lemma 3, this means that $H_\Delta$ will also have finite VC so the learnability of $H$ implies the learnability of $H_\Delta$.
> In light of your question, we will add a more detailed explanation of the difficulties of finding a general bound as well as our new result above to the final version of the paper.
>
> **״Could you comment on how the results in section 6 help the learner to choose model classes in practice?״**
> Certainly, here are some examples:
> 1. One implication of our results is that strategic behavior can generally either increase or decrease the complexity of the chosen model class. However, Thm. 2 ensures that VC can *only increase* for any class that is closed to input scaling, which includes many common classes used in practice. Thus, a learner that chooses such classes is guaranteed that the potential expressivity of the learned classifier will not be deteriorated by strategic behavior.
> 2. Prop. 4, Thm.3, and our new result detailed above all give upper bounds on the SVC of particular classes (e.g., ReLU neural networks). The implication is that a learner choosing work with such H is guaranteed that if H is learnable, then the induced $H_\Delta$ is also learnable.
> 3. Obs. 5 and Thm. 1 both point to the fact that several non-strategic classifiers $h$ can be mapped to the exact same effective classifier $h_\Delta$. If a class $H$ includes many such cases, then it has significant redundancy, which can have negative implications on the process of choosing a good $h \in H$ via learning (e.g., in terms of generalization, optimization, or the effectiveness of proxy losses). Ideally the learner should be cautious of classes in which this is prevalent, for example classes that permit high curvature.
>
> As noted, we will add these conclusions and others in a new section in the next revision.
>
> **”The experimental results in section 7.2 are also somewhat unsurprising: showing improved accuracy with bigger separation between positive and negative examples and smaller manipulation radius, with H^{lin} being a lower bound.”**
>
> The idea of this experiment was not to show how the maximum strategic accuracy improves with class separation and smaller manipulation radius, but rather to show how this trend compares with the non-strategic and linear baselines. This experiment ultimately demonstrates two things:
>
> 1. That the limited accuracy phenomenon detailed in Sec. 6.2 has a noticeable effect on practical cases and not just edge cases. This is evident by the noticeable deviation of $H_\Delta^* $ from the upper baseline of $H^*$.
> 2. That non-linearities in the strategic setting still greatly improve performance (which also serves as a justification for further study of non-linear classifiers in strategic classification). This is evident by the noticeable deviation of $H_\Delta^* $ from the upper baseline of $H^{lin}$.
>
> As such, the class separation and manipulation radius are varied not to point out an inferable trend, but rather to quantify both of the above conclusions.

---

> > ### Comment · Reviewer_qnCe · 2025-08-04
> >
> > Thank you for the detailed response. I have a follow-up question regarding the model-class level results: do Theorems 2 and 3 rely on the input domain being unbounded? Would the results still hold if the input feature vectors are restricted to have bounded norm, which seems like a more realistic assumption than allowing them to scale arbitrarily?

---

> > > ### Author Response · Authors · 2025-08-04
> > >
> > > Thank you for your question and for taking the time to read over our response. The high-level answer is that our results *do not* require an unbounded input space, but this is somewhat nuanced, so allow us to elaborate. Generally in strategic classification, if we assume that raw inputs $x$ are bounded by say $\|x\| \le B$, then for the problem to be well-defined we must allow this space to expand at the minimum to $B+\alpha$ to accommodate for strategic input modifications. More generally, we must ensure that $\|\Delta_h(x)\| \le B(\alpha)$ holds for all $x$ and any $h$, for some expansion $B(\alpha)>B$, where the exact form of $B(\alpha)$ depends both on the cost function and on $H$.
> > >
> > > Thm. 3 does not depend on scale (as long as the minimal expansion $B(\alpha) = B+\alpha$ is satisfied, as in all results). Note Thm. 3 applies to a particular class $H$.
> > >
> > > Thm. 2 does not depend on scale in that for any given learnable $H$, there exists some finite $B(\alpha)$ for which the result holds. In that sense, the result does not require an arbitrary expansion of the input space, only a finite one. However, since the statement is generic (i.e., applies to all learnable $H$ closed to scaling), we do not have a closed-form expression for $B(\alpha)$ that applies simultaneously to all such $H$.
> > >
> > > We hope this helped address your question and would be happy to answer any others you may have.

---

> > > > ### Comment · Reviewer_qnCe · 2025-08-08
> > > >
> > > > Thank you for the explanation. I appreciate the new result on piecewise linear functions and the discussion on how to choose model classes under strategic manipulation. I am happy to raise my score to 4.

---

### Official Review · Reviewer_UkjG · 2025-07-03

**Clarity:** 3
**Significance:** 2
**Originality:** 2
**Rating:** 3
**Confidence:** 3

**Summary:**

This paper studies strategic classification, where agents can intentionally modify their input features (while incurring an L-2 cost) to receive favorable outcomes from a classifier (e.g. in credit approval). This paper explores the implications of non-linear classifiers in such a setting.

Thanks to the assumption that the L-2 cost is the same for all agents, Sec 4, 5 and 6.2 effectively reduces the problem to study the behavior of offset curves/surfaces in the context of classification.

Sec 6.1 studies the behavior of offset curves in the language of VC dimension.

Edit: Author's response to reviewer YJeu has made me feel more positive about the paper. I'm raising my score to a 3.

**Questions:**

See weaknesses

**Ethical Concerns:**

["NO or VERY MINOR ethics concerns only"]

**Final Justification:**

Author's response to reviewer YJeu has made me feel more positive about the paper. I'm raising my score to a 4. Overall I still feel negative about the paper, as there is a lack of technical insight besides those derived from offset curves. The claimed application to the practical learning is questionable as well. I am unsure how they can be implemented.

**Limitations:**

Yes

**Quality:**

3

**Strengths And Weaknesses:**

Pros:

The paper is extremely well-written, with intuitions and/or illustrations provided for almost every proposition and proof. The analysis is well-structured and engaging.

Cons:

The paper makes connection to the study of offset curves, yet largely dismiss it as it "does not convey direct implications for learning". However, I feel that the paper could have used more direct connection to and terminology from offset curves, e.g. self-intersections, tangent vectors, instead of reinventing terminology. It is a connection that should be highlighted and celebrated.

Every claim that is made up to (and include) Theorem 1 is somewhat immediate once the reader understand the basic mechanism of offset curves.

---

> ### Author Rebuttal · Authors · 2025-07-31
>
> Thank you for your constructive review, we appreciate your expertise and point of view. We would however like to take the opportunity and present a broader perspective for why we believe our paper can be of interest to the community. The contribution of our paper lies not in any particular result or technique, but in the overall story it tells of strategic learning in the non-linear setting. Our goal was to raise awareness to the non-trivial implications of non-linearity which, by focusing predominantly on the linear case, the community has so far mostly assumed away (surveying many papers on strategic classification and their reviews, it was rare to find a paper which was *not* scrutinized for the exclusive focus on linear classifiers). We do not feel that this aim necessitates complex claims but rather a strong foundational understanding of the setting, which we believe our paper delivers. By the request of other reviewers, we will add a section on practical implications and take-aways of our results, which should promote our original goal and therefore strengthen the paper (for a detailed description please see our responses to Rev. YJeu and Rev. qnCe). We therefore kindly ask that you be willing to reconsider our paper, and whether it makes a sufficient contribution to the strategic learning literature, under this perspective.
>
> We now turn to address your particular concerns in detail.
>
> **”The paper makes connection to the study of offset curves, yet largely dismiss it as it
> ‘does not convey direct implications for learning’”**
> Please note that our intent was not to *dismiss* the connection – we did not mean for this to be implied in our writing, and apologize if this was the case. Rather, our intent was to depict offset curves as a useful idea for reasoning about certain aspects of strategic learning that proved useful in the context of our work. While we agree that the connection might extend beyond what we have presented (and that this is an interesting avenue to pursue), our targeted focus was intentional and aimed to support our own goals.
>
> In particular, and as Reviewer qnCe notes, our paper’s primary contribution is the class-level analysis in Sec. 6. To the best of our knowledge, such analysis lies outside the domain of offset curve theory, which concerns mostly point- and curve-level properties. We are unaware of results that connect curvature to the complexity of a function class or its VC dimension (nor do we believe that such connections are trivial). This is what we meant by “does not convey direct implications for learning”. Additionally, while our low-level analysis begins with focus on the L2-norm cost, both because it is the most popular choice in the literature, and because we found it to offer the best intuition, most of our results generalize beyond the L2 nor cost (please see our response to Rev. 4VxQ).
>
>
> **”I feel that the paper could have used more direct connection to and terminology from offset curves, e.g. self-intersections, tangent vectors, instead of reinventing terminology.”**
> While our results make use of ideas from offset curves, we do not see why they require a more direct connection than presented; our bottom-up construction ensures that we make use of the minimal set of notions necessary for our purposes. However, if this turned out to appear as “reinventing terminology”, then this was entirely unintentional, and we would be happy to make the required corrections and adjustments. Please let us know if you have any specific requests or suggestions and we will gladly accommodate them in the next revision.
>
> **”Every claim that is made up to (and include) Theorem 1 is somewhat immediate once the reader understand the basic mechanism of offset curves.”**
> Secs. 4 and 5 are intended to build intuition and a strong foundation towards Sec. 6. With the common reader in mind, we felt that a detailed exposition was imperative – even if this resulted in a somewhat easy read for the more familiar audience.
> However, and while our earlier results are indeed observations (and are stated as such), we respectfully disagree that all results that follow are “immediate”. An observation or result that is simple (in hindsight) does not mean it cannot be useful to the community. In fact, we view it as a strength that our paper makes use of simple observations to derive a meaningful description of a complex learning task. Given that this showcases the potential for using offset curves for this purpose, we are unsure why you view this as a limitation.
> The contribution of our work lies not in any individual claim but in the story it tells of non-linear strategic learning as a whole: for example, that strategic learning cannot be universal, that class complexity can either increase or decrease, and that some points have no effect on learning outcomes. We believe such conclusions can be of interest to the strategic learning community, and of value towards future work.

---

### Official Review · Reviewer_4VxQ · 2025-07-05

**Clarity:** 3
**Significance:** 3
**Originality:** 3
**Rating:** 4
**Confidence:** 3

**Summary:**

This paper investigates strategic classification under non-linear models, extending prior work that largely focuses on linear classifiers. In strategic settings, users manipulate their input features to receive favorable outcomes, which fundamentally alters the learning problem. The authors make a bottom-up analytical and empirical approach to study how non-linear decision boundaries, classifier expressivity, and model class complexity behave in strategic environments. An interesting insight is that universal approximators like neural networks lose their universality when faced with strategic agents. The authors support their analysis with theoretical characterizations.

**Questions:**

1. Could the non-universality of neural networks under strategic manipulation be quantified more rigorously?

2. How sensitive are your findings to the assumption of optimal best-response from agents? Would bounded rationality change the picture?

**Ethical Concerns:**

["NO or VERY MINOR ethics concerns only"]

**Final Justification:**

The rebuttal addresses my concerns. I recommend the acceptance.

**Limitations:**

See weakness

**Quality:**

3

**Strengths And Weaknesses:**

**Strengths**

1. Strategic classification is relevant in high-stakes applications, and extending its understanding beyond linear classifiers is both novel and necessary.

2. The insight that universal approximators are no longer universal under strategic manipulation is intellectually significant and well-motivated.

3. The paper rigorously analyzes the effects of strategic behavior at three levels: individual inputs, decision boundaries, and model classes.

4. The exposition is clear and logical. Definitions, examples, and motivations are carefully laid out.

**Weaknesses**


1. The analysis focuses primarily on $l_2$ cost; broader conclusions might require considering other realistic cost structures (e.g., non-Euclidean, asymmetric, feature-dependent).

2. While the claim that neural networks are no longer universal is well-motivated and empirically observed, a more formal characterization  would strengthen the main claim.

3. The paper focuses on theoretical behavior but gives limited discussion on how this insight could influence real-world tasks in strategic settings.

---

> ### Author Rebuttal · Authors · 2025-07-31
>
> Thank you for your positive review and useful comments, we appreciate the effort and time spent in providing us with helpful feedback. Please see our detailed response and answers to your questions below.
>
> **”The analysis focuses primarily on L2 cost; broader conclusions might require considering other realistic cost structures (e.g., non-Euclidean, asymmetric, feature-dependent).”**
>
> Indeed, our low-level analysis begins with focus on the L2-norm cost, both because it is the most popular choice in the literature, and because we found it to offer the best intuition. Nonetheless, **most of our results hold more broadly**, and in particular, all theorems and propositions in Sec. 6 generalize to any $ L_p $ norm:
>
> 1. Prop. 4 applies generally, since negative-inside polytopes are still mapped to smaller negative-inside polytopes under general $L_p$ cost functions.
> 2. Thm. 1, Cor. 1, and the example in Figure 4 of increasing VC dimension remain under minor tweaks to the constructions that take into account the differences in shapes  of $L_p$ balls for different values of $p \ge 1$.
> 3. Thm. 2 holds for any cost function under which the maximum distance any user can strategically move is bounded by a constant. This holds not only for $L_p$ norms but also for most reasonable cost functions.
> 4. Thm. 3 is stated for $L_1, L_2$, and $L_\infty$. However, a more general result can be stated with dependence on the VC dimension of the cost function artifacts (i.e. the set C = { $c(x,x’) \leq \alpha \\ \\ \forall x \in R^d $ }  ). The bound would then be $O(kd^2 \\ logk + VC(C))$, and would support any plug-in results for $VC(C)$.
> 5. Prop. 5, and Prop. 6 also  remain under minor tweaks to the constructions that take into account the differences in shapes  of $L_p$ balls for different values of $p \ge 1$.
> 6. Obs. 3-5, Cor. 2, hold for $L_p$ norms in general as well since they are not shape dependent, but rather results of movement within a ball.
>
> Thus, our claims regarding the fact that non-learnable classes can become learnable, the loss of universality, and the limits to approximation all hold more generally.
>
> In terms of asymmetric norms such as the Mahalanobis norm, while these norms would complicate the curvature analysis (and subsequent impossibility results that build off them), all results in Sec. 6 would still hold. This is because while the shape of balls changes (like with other $L_p$ balls), fundamental properties like lines mapping to other lines do not. As such, the results in Sec. 6 would still hold for the exact same reasons as delineated above for $L_p$ balls.
>
> As for feature-dependent (also known as “instance-wise”) costs, these are qualitatively different from the conventional global costs, in a way which can make the claims become irrelevant or even degenerate. For example, Sundaram et al. [2021] show that on the one hand, even for linear $H$, instance-wise cost function can cause $VC(H_\Delta) = \infty$, and on the other hand, for separable costs (which are instance-wise) it holds that $VC(H_\Delta) \le 2$ for *any* $H$.
>
>
> **”While the claim that neural networks are no longer universal is well-motivated and empirically observed, a more formal characterization would strengthen the main claim.”**
> Thank you for this suggestion. Our notion of non-universality states that there exists a classifier $h$ for which no effective class $H_\Delta$ can include $h$ as a member. We will gladly add this to the paper.
>
> **”Could the non-universality of neural networks under strategic manipulation be quantified more rigorously?”**
> This is an interesting idea. While we chose to work with a rigid definition, it is also possible to define (non)-universality in a more quantifiable manner based on approximation, as is common in the expressivity literature. However, and to the best of our knowledge, this literature focuses predominantly on regression, and approximation is defined with respect to the actual values of the functions considered (typically w.r.t. the infinity norm). Such definitions therefore do not carry over to classification where values are binary, and we are unaware of any standard notion of universality for decision boundaries. One suggestion for an adaptation of the existing definition to the classification setting is the following:
>
> Let $X$ be some input domain, and let $\Omega =$ {$g : X \to R$ } be a collection of scalar functions on $X$. Define $h_g(x)=sign(g(x))$. Let $F=${$f : X \to R$} be a class of scalar functions, and define the corresponding hypothesis class as $H=H_F=${$h_f:f \in F$}. Then $H$ is a universal approximator w.r.t. $\Omega$ if the following holds:
>
> $\forall g \in \Omega \\ \\ \forall \epsilon>0 \\ \\ \exists f \in F$ s.t. $h_f(x)=h_g(x) \\ \\ \forall x \in X$ with $|g(x)|>\epsilon$.
>
>
> Intuitively, this states that $H$ is universal if for every ground truth $h^* = h_g$ there is some $h=h_f \in H$ that agrees with $h^* $ on all inputs $x$, except possibly on those that are $\epsilon$ - close to the decision boundary of $h^*$, for arbitrarily small $\epsilon$.
>
>
> **“The paper focuses on theoretical behavior but gives limited discussion on how this insight could influence real-world tasks in strategic settings.”**
> Thank you for pointing this out. Given this and similar comments by other reviewers, **we will add a section on practical take-aways from our results in the next revision using the extra page.** This section will include:
> * General guidelines for a potential inversion-based learning approach, with benefits and challenges
> * An idea for regularizing based on curvature
> * A method for computing the interpolation threshold, which in a strategic setting can occur at strictly positive training error
> * Insights on how to choose model classes, and which classes are best avoided
> * Observations on how to set expectations regarding sample size, possible expressivity, and attainable strategic accuracy
> For a more detailed description of these points please see our responses to Rev. YJeu and Rev. qnCe.
>
>
> **“How sensitive are your findings to the assumption of optimal best-response from agents? Would bounded rationality change the picture?.”**
> This is an interesting question. The answer likely depends on how rationality is broken. For example, if users are simply loss averse, then the shape of the cost is retained, and all results should still apply. We believe this would hold also for users that operate under partial information; for example, if they are Bayesian, then as long as the prior is smooth, the posterior should also be smooth, and outcomes will be similar. If however users are computationally bounded (i.e., cannot solve the argmax in $\Delta$ exactly), then this can be more difficult to model because it requires understanding how their approximation of $\Delta_h(x)$ alters the shape of the cost function artifact. Some cases might be tractable, for example if users move according to the gradient of an underlying score function $f(x)$ (where $h(x)=sign(f(x))$) or if they move w.r.t. a local linear approximation of the decision boundary of $h$. Other less-structured cases however are likely to be much more challenging to work with.

---

> > ### Comment · Reviewer_4VxQ · 2025-08-05
> >
> > I am satisfied with the authors’ clarifications. I look forward to seeing the revised version with the promised additions and clarifications.

---

### Official Review · Reviewer_YJeu · 2025-07-06

**Clarity:** 2
**Significance:** 2
**Originality:** 3
**Rating:** 4
**Confidence:** 4

**Summary:**

They study Strategic Classification (SC) in a non-linear setting. Unlike SC in a linear setting, when the decision boundary is non-linear, datapoints move in different directions, and hence the effective decision boundary would get distorted. Their results are in 3 different categories: 1)points, 2)classifiers 3)model classes in a non-linear setting.

Section 4 describes what would happen to points, how a point on h gets mapped to corresponding points in the effective decision boundary h_{\delta}, cases 1-4 describe 4 possibilities in this case, there might be a 1-1 mapping, wipeout, expansion, or collission.

Section 5 demonstrates what would happen to individual classifiers, in terms of their curvature and containment (number of positive regions in some sense).

Section 6 explains what would happen to the VC dimension of the model classes; they might increase or decrease.

**Questions:**

Eq 1, shouldn't it be h(x')?

Lines 114 - 117, confusing since you are using x as a modified point in line 114 and then as an original point in line 117.


Figure 2 needs more explanation. I do not understand what all the colors represent. It seems like you are addressing 4 categories, but there
are 5 figures. It is very hard to understand what is going on.

Obs3 for what points x? The ones on the decision boundary?

Confused about the wording of section 6.1. Do you mean that in some cases, the  VC-dimension of the effective classifier sometimes decreases, and sometimes it increases?


What does line 292 mean?  For what point x?

In your experiments, how do you compute the effective decision boundary?

**Ethical Concerns:**

["NO or VERY MINOR ethics concerns only"]

**Final Justification:**

I agree that learning in a non-linear setting is quite challenging; however, after going through other responses, I decided to keep my score.

**Limitations:**

Although it is very important to study SC in non-linear setting, my biggest complain about this work is that it is mostly structural results, it does not give any algorithms for learning in non-linear settings.

**Paper Formatting Concerns:**

I do not have any specific formatting concerns.

**Quality:**

3

**Strengths And Weaknesses:**

Although it is very important to study SC in non-linear setting, my biggest complain about this work is that it is mostly structural results, it does not give any algorithms for learning in non-linear settings.

---

> ### Author Rebuttal · Authors · 2025-07-31
>
> Thank you for your careful review and detailed comments. Please see our response below.
>
> **"[The work] does not give any algorithms for learning in non-linear settings.”**
> We certainly agree that this would give added value, but note that optimization in strategic batch learning is quite challenging even in simpler settings, and so the design of a principled algorithm for the general non-linear case is a difficult ask. Existing works rely on strong assumptions to enable optimization. For example, the original paper of Hardt et al. [2016] provides an algorithm, but requires assumptions on the cost function which reduce the problem to a uni-dimensional learning task, and their algorithm is essentially a line search. Levanon and Rosenfeld [2021] differentiate through $\Delta$, but this relies strongly on this operation being a linear projection – which applies only to linear classifiers. Neither of the above naturally extend to the non-linear case. Levanon and Rosenfeld [2022] propose the generalized strategic hinge, which does apply to non-linear classifiers, but requires solving a generally intractable nested argmin objective (which can be even harder than solving the original nested argmax objective).
>
> Given this difficulty, our paper aims to first establish an understanding of the challenges inherent to non-linear strategic learning, and in particular in relation to the linear case. We hope our results and conclusions can help guide the future design of learning algorithms, as well as set expectations for what can be achievable, and what cannot.
>
> Towards this, and motivated by your review, **we provide here some examples of how our results can be used as practical take-aways for effective learning**, which we will add to the paper as a new section and with further discussion using the extra page:
>
> 1. **Learning via inversion**: One implication of our function-level analysis is that for any classifier $h$ with strategic accuracy $s-acc(h)$, any effective classifier $h’$ s.t. $h \mapsto h_\Delta=h’$ has regular (i.e., non-strategic) accuracy $acc(h’) = s-acc(h)$. Thus, one approach to optimizing strategic accuracy is via reduction: (i) train $h’ \in H’$ to maximize *non-strategic* accuracy using any conventional approach for some choice of $H’$, and then (ii) apply the *inverse* function mapping (which may not be 1-to-1) to obtain a non-strategic classifier $h$, whose effective decision boundary is that of $h’$. This approach requires the ability to solve the “inverse” problem (which may not be 1-to-1) to find the effective classifier. We discuss cases where this would and wouldn’t work in Sec. 5.3.
>
> 2. **Regularizing curvature**: A sufficient condition for a pre-image strategic $h’$ to exist for a given non-strategic $h$ is that the function mapping is 1-to-1. Rather than restrict learning to only invertible classes $H’$ a-priori, an alternative approach is to work with general $H’$ but regularize against those $h’ \in H’$ which do not have an inverse. Prop. 1 implies that one way to promote this is by promoting the smoothness of $h’$, since low-curvature classifiers are less prone to direct wipe-out, and therefore more likely to permit inversion.
>
> 3. **Finding the interpolation threshold**: The interpolation threshold – the point in which the number of model parameters (or model complexity more generally) attains minimal training error – is key for learning in practice: it marks the extremal point of overfitting (in the classic underparametrized regime) and the beginning of benign overfitting (in the modern overparametrized regime, e.g., of deep neural nets). In non-strategic learning, this point is easily identified as that in which the training error is zero. In contrast, our results show that in strategic learning, the minimal training error can be strictly positive. This makes it unclear when (and if) the threshold has been reached. Fortunately, our procedure for our approximation experiment in Sec. 7.2 can be used generally to compute the minimal attainable strategic training error, independently of the chosen model class. This provides a tool that can be used in practice to measure interpolation.
>
> Please also see our response to Rev. qnCe regarding practical implications of our results in Sec. 6, which we also plan to incorporate into the new section on take-aways. We believe that these additions will strengthen our paper – thank you for this suggestion!
>
>
>
> **Minors:**
>
> > Eq 1, shouldn't it be h(x')?
>
> Yes, that is correct. We found this mistake a few days after the deadline and have since corrected it.
>
> >Lines 114 - 117, confusing since you are using x as a modified point in line 114 and then as an original point in line 117.
>
> Our intention was to use x in line 114 to represent the points on the original decision boundary, but we see where this might create confusion so we will update the notation.
>
>
> > Figure 2 needs more explanation.
>
> Apologies for the confusion. There are indeed four categories, but the wipeout category has two “sub-types”, giving a total of five cases. From left to right we present: one-to-one (case #1), direct wipeout (case #2a), indirect wipeout (case #2b), expansion (case #3), and collision (case #4). We will edit the caption to clarify this. In terms of colors, the original classifiers ($h$) are drawn in black and the effective classifiers ($h_\Delta$) are dashed and grey. We signify the positive and negative regions of the original classifier with blue and red respectively.
>
> > Obs3 for what points x? The ones on the decision boundary?
>
> No, this is talking about entire contained regions of points including those not on the decision boundary. Any negative region C in h that is fully contained in some $\alpha$-ball centered at some x will become positive in $h_\Delta$.
>
> > Confused about the wording of section 6.1. Do you mean that in some cases, the VC-dimension of the effective classifier sometimes decreases, and sometimes it increases?
>
> Yes, the VC dimension may increase or decrease.
>
> >What does line 292 mean? For what point x?
>
> The intention is for any arbitrary x. The classifier is one that classifies some radius 0.9α ball centered at some x as positive and is negative elsewhere.
>
>
> >In your experiments, how do you compute the effective decision boundary?
>
> The effective decision boundary was only computed in the first experiment. Because we are concerned specifically with the polynomial fit of $h_\Delta$, we label the points in a fine grid G over $[-100, 100]^2$ both regularly and then strategically and use the strategic labels to get a polynomial fit. In this way we don’t directly compute the effective classifier (which would be difficult), but approximate it well enough to get the polynomial fit as well as visualize it.

---

> > ### Comment · Reviewer_YJeu · 2025-08-04
> >
> > Thank you for your response. I agree that adding these points in a new section would give a better picture of the importance of your work. Also, I agree that learning in a non-linear setting is quite challenging. I don't have any further questions right now.

---

### Decision · Program_Chairs · 2025-09-17

**Decision:**

Accept (poster)

**Comment:**

This paper investigates strategic classification in non-linear settings, extending prior work that has largely focused on linear models. In strategic classification, agents deliberately modify their input features, typically incurring an l2 cos to obtain favorable outcomes (e.g., in credit approval). Unlike the linear case, where all points shift in the same direction, non-linear decision boundaries induce heterogeneous movements of datapoints, leading to distortions in the effective boundary. The authors take a bottom-up analytical and empirical approach to examine the implications of non-linearity across three levels: individual points, classifiers, and model classes. A key insight is that universal approximators such as neural networks lose their universality when faced with strategically responding agents. Their findings are supported by theoretical characterizations and empirical analysis.

The reviewers agreed that this non-linear extension and the technical results are exciting. Though most of the presented results are structured results, I believe this result will motivate future studies and discussions on this important topic.